# *Sketch-Plan-Generalize*: Learning Inductive Representations for Grounded Spatial Concepts

## Abstract

Our goal is to enable embodied agents to learn inductive representations for grounded spatial concepts, e.g., learning staircase as an inductive composition of towers of increasing height. Given few human demonstrations, we seek a learning architecture that infers a succinct inductive *program* representation that *explains* the observed instances. The approach should generalize to learning of novel structures of different size or complexity expressed as a hierarchical composition of previously learned concepts. Existing approaches that use code generation capabilities of pre-trained large (visual) language models as well as purely neural models show poor generalization to *a-priori* unseen complex concepts. Our key insight is to factor inductive concept learning as: (i) *Sketch:* detecting and inferring a coarse signature of a new concept (ii) *Plan:* performing MCTS search over grounded action sequences (iii) *Generalize:* abstracting out grounded plans as inductive programs. Our pipeline facilitates generalization and modular re-use enabling continual concept learning. Our approach combines the benefits of code generation ability of large language models (LLMs) along with grounded neural representations, resulting in neuro-symbolic programs that show stronger inductive generalization on the task of constructing complex structures vis-á-vis LLM-only and purely neural approaches. Further, we demonstrate reasoning and planning capabilities with learned concepts for embodied instruction following.

## 1 Introduction

The ability to learn *inductive* representation for novel grounded concepts is one of the hallmarks of human intelligence (Tenenbaum et al., 2011). Humans are highly data efficient – observing a few instances of *towers* of a certain heights, we can generalize to constructing *towers* of *any* height. Further, we interpret increasingly complex concepts as hierarchical composition over simpler ones, e.g., a *tower* as a sequence of blocks placed on top of each other, or a *staircase* composed of towers of increasing height. This paper considers the problem of learning a program representation, from a few demonstrations, that models the *inductive* realization of grounded spatial concepts. Learning of such concepts is a challenging task due to an expansive space of programs and the need to reason about their physical plausibility. Further, the representation must support inductive generalization over learned concepts as well as express complex hierarchical concepts via modular re-use of concepts learnt previously.

Prior efforts such as Liang et al. (2023) uses a LLM (Large Language Model) to generate control program for a given task specification but fail to generalize to complex spatial concepts which are difficult to tokenize. Extension of this to VLMs (Vision Language Models), Achiam et al. (2023), also fail to generalize when presented with linguistically novel concepts, due to over reliance on prior knowledge and their inability to effectively learn novel concepts from given demonstrations. On the other hand, neural approaches such as Liu et al. (2023), learn from given demonstrations but generalize poorly due to their inability to (a) explicitly model symbolic concept of induction and (b) modularize as well as re-use previously acquired concepts. Approaches such as Li et al. (2019) train RL-based policies to attain spatial assembly by encoding an inductive spatial prior using GNNs within the policy architecture. However, generalization is still limited and assumes an elicitation of the goal as per object positions, resulting in lack of ability to take in goal description such as "construct a tower of size three". In essence, we attribute poor generalization of such approaches to an *implicit entangling* of the following objectives: (i) postulating a high-level program for new concepts,

Figure 1: **Problem Overview.** Our goal is to enable an embodied agent to learn grounded and generalizeable representations for spatial abstractions possessing a notion of induction (e.g., constructing a tower, row or their combinations such as staircases, boundary etc.). Learning is enabled by querying prior knowledge from large pre-trained models, performing search in the action space guided by observations of a human demonstration for few examples and finally generalizing as compact programs. (Left) A human demonstrates the construction of a *row* and *tower* of size three. (Right) The agent learns *program* representation that enables inductive generalization to novel structures (varied sizes and visual attributes) and expresses complex concepts as hierarchical composition of previously acquired ones. E.g., learning a *tower* as a sequence of blocks placed one on top of another and a *pyramid* as rows of decreasing size.

(ii) evaluating plausibility of grounded plans to align with human demonstration of concepts to be learned and (iii) abstracting out a grounded program to facilitate inductive generalization and modular re-use in a continual manner.

This paper introduces an approach, termed *SPG*, that *factorizes* the concept learning task as: (a) *Sketch*: Given a language-annotated demonstration of a novel concept, an LLM is used to postulate a function signature. (b) *Plan*: Refinement of program sketches via MCTS search, rapidly evaluating actions sequences guided by a reward associated with constructing a concept. The search is accelerated by training a neural action predictor that uses the given demonstrations. (c) *Generalize*: Leveraging the code generation capability of an LLM to distill grounded plans into a program that is inductively generalize-able. This results in a continually evolving library of concepts which can be used to hierarchically learn complex concepts in future. The modular architecture enables continual learning by providing the ability to decide whether the new concept encountered either as a symbolic composition of existing concepts, or, a neural embedding trained via gradient update. Our experiments demonstrate accurate learning of simple and complex concepts from few demonstrations for a range of spatial structures. Further, our approach shows inductive generalization in out-of-distribution settings, significantly improving over the baselines. We also present deeper insights around the efficiency gains obtained by combining symbolic MCTS with neural action predictor. Finally, we show how learned concepts can be grounded in the visual input, enabling a robot to follow natural language instructions referring to *a-priori* unseen spatial configurations.

## 2 RELATED WORKS

**Concept Learning:** The problem of acquiring higher-order programmatic constructs is often modeled as Bayesian inference over a latent symbol space given observed instances. Seminal works have demonstrated efficient inference over latent generative programs to express hand-written digits (Lake et al., 2015), object arrangements (Ellis et al., 2018), motion plans (Mao et al., 2019), goal-directed policies (Silver et al., 2019) or compressed/refactored code (Grand et al., 2023; Ellis et al., 2021). These works are focused on learning abstract programs without considering their grounding in the 3D world or the process of constructing them (e.g., via an embodied agent performing stacking). In contrast, this paper focus on learning a representation of specific class of higher-order spatially-grounded concepts, namely those possessing a notion of induction resulting in the construction of a structure. While prior efforts have leveraged program synthesis/search methods in learning concepts, we expose such a search to the assessing the physical construction plausibility thereby learning physically grounded concepts.

**Learning-to-plan Methods:** Our work is complementary to efforts that learn symbolic constructs for efficient planning. Works such as Silver et al. (2023; 2024); Liu et al. (2024), infer state-action abstractions for planning by querying large pre-trained models or by optimizing a goal attainability objective. This paper, instead, focuses on learning a representation for complex spatial assemblies as inductive programs leading to the ability to infer complex goal specifications which can then be combined with aforementioned works for synthesize efficient plans to realize complex assemblies. Works such as Li et al. (2019) learn to construct structures by encoding relational knowledge via graph neural network. However, this effort suffers from poor generalization to unseen examples, (e.g., tower of larger size) and do not possess a mechanism to re-use previously acquired concepts. Works such as Wang et al. (2023a;d) shows lifelong learning of skills by learning to plan high-level tasks through composition of simple skills for simulated agents. Others (Wan et al., 2023; Parakh et al., 2023) initiate new skill acquisition upon detecting task failure, building a library of skills over time. However, they do not model deep inductive use of learned concepts and initiate skill acquisition only upon failure as opposed to learning continually even from goal-reaching demonstrations.

**Robot Instruction Following:** Instruction following involves grounding symbolic constructs expressed in language with aspects of the state-action space such as object assemblies (Paul et al., 2018; Collins et al., 2024; Lachmy et al., 2022), spatial relations (Tellex et al., 2011; Kim et al., 2024), reward functions (Boularias et al., 2015), or motion constraints (Howard et al., 2014). These works assume the presence of grounded representation for symbolic concepts and only learn associations between language and concepts. In contrast our work *jointly* learns higher-order concepts composed of simpler concepts along with their grounding in the robot's state and action space. Others (Singh et al., 2022; Wang et al., 2023b; Ahn et al., 2022; Liang et al., 2023) leverage prior-knowledge embodied in large vision-language models to directly translate high-level tasks to robot control programs. Our experiments (reported subsequently) demonstrate their limitation in outputting programs for structure assembly-type tasks that require long-range (inductive) spatial reasoning and consideration of physical plausability of construction. Our approach addresses this problem by *coupling* abstract task knowledge from pre-trained models with physical reasoning in the space of executable plans.

# 3 PRELIMINARIES AND PROBLEM SETTING

We consider an embodied agent that uses a visual and depth sensor to observe its environment and can grasp and release objects at specified poses. We represent the robot's domain as a goal-conditioned MDP $< \mathcal{S}, \mathcal{A}, \mathcal{T}, g, \mathcal{R}, \gamma >$ where $\mathcal{S}$ is the state space, $\mathcal{A}$ is the action space, $\mathcal{T}$ is the transition function, $g$ is the goal, $R$ is the reward model and $\gamma$ is the discount factor. The agent's objective is to learn a policy that generates a sequence of actions from an initial state $s_0$ to achieve the goal $g$ in response to an instruction $\Lambda$ specifying the intended goal from a human. We assume that the agent possesses a model of semantic relations (e.g., left(), right() etc.) as well as semantic actions such as moving an object by grasping and releasing at a target location. Such modular and composable notions can be acquired from demonstrations via approaches outlined in Kalithasan et al. (2023); Mao et al. (2019; 2022). Such notions populate a library of concepts $\mathcal{L}$ available as grounded executable function calls. Following recent efforts (Liang et al., 2023; Huang et al., 2022; Ahn et al., 2022; Singh et al., 2022) in representing robot control directly as executable programs, we represent action sequence corresponding to a plan as a program consisting of function calls to executable actions and grounded spatial reasoning.

Our goal is to enable a robot to interpret and learn the concepts in instructions such as *"construct a tower with red blocks of height five"*. Specifically, we aim to learn spatial constructs like a tower that requires sequential actions that repeatedly place a block on top of a previously constructed assembly, a process akin to induction. Given a few demonstrations of constructing a spatial assembly, $\mathcal{D}$, each consisting of natural language description $\Lambda$ ("construct a tower of red blocks of size five") and a sequence of key frame states $\{S_1, \cdots, S_g\}$ associated with the construction process, we seek to learn a program that models the inductive nature of the concept of tower. This learned representation should enable the agent to generalize inductively to new instructions, such as "construct a tower of blue blocks of height ten." Moreover, the learned concepts should facilitate the learning of more complex structures, which are challenging to represent using primitive actions alone. For example, the concept of a "tower" should assist in learning a "staircase," which can be represented as a sequence of towers of increasing heights.

## 4 REPRESENTING INDUCTIVE SPATIAL CONCEPTS

We formalize the notion of inductive spatial concepts and formulate the learning objective.

**Inductive Spatial Concepts:** A spatial structure is an inductive concept if its construction can be described recursively using a similar structure of smaller size or as a composition of other simpler structures. Formally, let $C_1, \cdots, C_{|\mathcal{L}|}$ represent the concepts in the concept library $\mathcal{L}$. We define a partial order on $\mathcal{L}$ where a concept $C$ is "dependent on" $\widetilde{C}$ if $\widetilde{C}$ is a substructure of $C$. For example, a staircase is dependent on a tower, and $X$ (cross) is dependent on diagonals, and so on. This partial order is referred to as structural complexity, where a concept $C$ is more structurally complex than $\widetilde{C}$ if $C$ is dependent on $\widetilde{C}$. Without loss of generality, assume that $C_1, \cdots, C_{|\mathcal{L}|}$ are written in topological order as per their structural complexity. Now, the construction of an inductive spatial concept $C_K$ of size $n$ at a position $p$, denoted by the function $h(C_k, n, p)$, is defined recursively as:

$$h(C_k, n, p) = \underbrace{h^\lambda(C_k, n-1, pos(.))}_{\text{Induction (I)}} \circ \underbrace{\prod_{l=1}^{L(C_k)} h\big(C_{k'_l}, \texttt{size}(.), pos(.)\big)}_{\text{Composition (C)}} \circ \underbrace{\prod_{l=1}^{L'(C_k)} \eta_\theta^l\big(pos(.)\big)}_{\text{Base (B)}} \quad (1)$$

where, $\lambda \in \{0, 1\}$, $k' < k$, $0 \le L(C_k), L'(C_k) \le o(|\mathcal{L}|)$, $pos(.) = pos(C_k, l, n, p)$ and $\texttt{size}(.) = \texttt{size}(C_k, l, n)$ are functions that predict the size and position of the structure to be constructed.

1. *Induction term:* The first term $h^\lambda(C_k, n-1, pos(.))$ is referred to as the induction term because it represents the possibility of constructing $C_k$ of size $n$ using $C_k$ of size $n-1$. Here, $\lambda$ is an integer exponent, either 0 or 1, where $\lambda = 0$ indicates the absence of the induction term, and $\lambda = 1$ indicates its presence.

2. *Composition term:* The second term $\prod_{l=1}^{L(C_k)} h\big(C_{k'_l}, \texttt{size}(.), pos(.)\big)$, called the composition term, allows us to express the construction of $C_k$ as a composition of previously known concepts in the library. The number of required compositions depends on the concept $C_k$ and the size of the library $\mathcal{L}$.

3. *Base term:* The third term $\prod_{l=1}^{L'(C_k)} \eta_\theta^l\big(pos(.)\big)$ defines the base case where the construction of concept $C_k$ may include $L'$ number of primitive actions. For example, the construction of a tower of size $n$ can be written as a construction of a tower of size $n-1$ followed by a primitive action of moving a block on top.

**Learning Objective:** The functional space of inductive concepts ($h$) leads to a hypothesis space $\mathcal{H}$ of associated neuro-symbolic programs. Each goal-reaching demonstration corresponds to a particular instantiation of a given inductive concept, i.e. $h(C_k, n, p)$, where the $p$ comes from the sequence of frames, and $n$, $C_k$ comes from $\Lambda$. We aim to learn a generic representation $H = h(C_k, \cdot, \cdot) \in \mathcal{H}$ for the given concept, which is general for all $n$ and $p$. Given (few) demonstrations of a human constructing a spatial structure, concept learning can be formulated as the Bayesian posterior (Lake et al., 2015; Shah et al., 2018; Silver et al., 2019), $\texttt{P}_\mathcal{H}(H|\Lambda, S_1..S_g) \propto \texttt{P}(S_1..S_g|\Lambda, H) \cdot \texttt{P}(H|\Lambda)$. Here, the likelihood term associates a candidate program, and the prior term regularizes the program space. The maximum *a-posteriori* estimate, representing the learnt program, is obtained by optimizing the following objective:

$$H^* = \arg\min_{H \in \mathcal{H}} \left[ \texttt{Loss}(\{S_1..S_g\}, Exec(H, \Lambda, S_1)) - \log \texttt{P}(H|\Lambda) \right] \quad (2)$$

Since exact inference is intractable, approximate inference is performed via search in the program space. Note that learning inductive spatial concepts given demonstration considers programs that represent plans that attain physically grounded/feasible structures, an object we model during the search. Additionally, we seek strong generalization from a few instances of an inductive structure to structures with arbitrary sizes, in effect favouring programs with iterative looping constructs.

Figure 2: **Method Overview.** We learn a neuro-symbolic program for inductive spatial concepts factored as (a) *Sketch* (b) *Plan* (c) *Generalize*. The example above shows the progressive realization of a program for the concept of a staircase acquired by observing a single demonstration of building a staircase of size four and its corresponding natural language instruction, *"construct a staircase of size four using magenta legos."*

## 5 LEARNING INDUCTIVE CONCEPTS FROM DEMONSTRATIONS

We address the problem of estimating a succinct generalized program, Eq. 2, modeling an inductive spatial concept modeling structures whose construction is observed in a human demonstration. Direct symbolic search in the space of programs is intractable (particularly due to looping constructs needed for modeling induction) but can explicitly reason over previously acquired concepts. Alternatively, neural methods attempting to predict action sequence to attain the assembly are challenged by continual setting where concepts can increase over time building on previously learnt ones but are resilient to noise. Our approach blends both approaches and factors the concept learning task as:

- **Sketch:** From the natural language instruction ($\Lambda$), we extract a task sketch ($H_S^*$) using an LLM that provides the signature (concept name and instantiated arguments) of the concept to be learned. When grounded in the initial scene of the demonstration, the task sketch provides a particular instance of the concept demonstrated in the given demonstration.
- **Plan:** MCTS-based search using the already learnt concepts that outputs the sequence of grounded actions, best explaining the given demonstration.
- **Generalize:** The grounded plan $H_P^*$ and task sketch $H_S^*$ are provided to an LLM to obtain a general Python program whose execution on the given scene matches the searched plan.

Formally, the factored exploration of the program space for a demonstration is performed as:

$$H_S^* \leftarrow Sketch(\Lambda\,;\,\theta_S)\,;\; H_P^* \leftarrow Plan(S_1..S_g, H_S^*;\,\theta_P)\,;\; H_G^* \leftarrow Generalize(H_P^*,\,H_S^*;\,\theta_G) \quad (3)$$

Here, $\theta_S$, $\theta_P$ and $\theta_G$ are the learnable parameters (including hyperparameters) of the Sketch, Plan and Generalize functions, respectively. The concept library $\mathcal{L}$ is initialized with primitive visual and action concepts. Upon acquiring a new inductive concept $H_G^* = H^*$, we update our library accordingly: $\mathcal{L} \leftarrow \mathcal{L} \cup H^*$. An example is provided in Appendix Sec. A.5. Fig. 2 illustrates an example of progressive prog. estimation. Next, we detail each of the three steps mentioned above.

### 5.1 (SKETCH) GROUNDED TASK SKETCH GENERATION

An LLM driven by in-context learning is used to get a program signature (a sketch) for a concept from the natural language instruction. The task sketch is a tree of nested function calls that outlines the function header (name and the parameters) of the inductive concept/program to be learned. A detailed exposition on prompting appears in the Appendix C.1. The task sketch thus obtained is then grounded on the input scene using a quasi-symbolic visual grounding module akin to Mao et al. (2019); Kalithasan et al. (2023); Wang et al. (2023c). This module has three key components: (1) a visual extractor (ResNet-34 based) that extracts the features of all objects in the scene, (2) a concept embedding module that learns disentangled representations for visual concepts like *green* and *dice*, and (3) a quasi-symbolic executor equipped with pre-defined behaviours such as "filter" to select/ground the objects of interest. For example, grounding the task sketch "Tower (height =3,

objects = filter(green, dice))" results in an instantiated function call "Tower(height = 3, objects = [1, 2, 3])" where [1, 2, 3] are the green coloured dice.

## 5.2  (PLAN) PHYSICAL REWARD GUIDED PLAN SEARCH

The plan search involves finding a generalizable plan that effectively explains the demonstration $S_1, \cdots, S_g$. Specifically, this involves determining the concepts, their respective grounded parameters, and the order of composition as specified in the Equation 1.

**Primitive Actions.** Constructing complex structures involves two steps: (1) identifying or imagining the placement location of an object/structure and (2) picking and placing the object at the imagined location. The position $pos_\theta(.)$ for placement is determined using a head, which represents a cuboidal enclosure in 3D space. Conceptually, moving the head is akin to the robot's cognitive exploration of potential placements to achieve the desired spatial configuration. We define a set of primitive functions, $\mathcal{A}_p$, to guide the movement and placement of objects in two stages: (1) `move_head(direction)`: This primitive moves the abstract head to a desired relative position and (2) `keep_at_head(objects)`: This primitive places the target object from the list `objects` at the current location of the head. It is important to note that `move_head(direction)` is a neural operator, which takes the head's current position and predicts its new location based on the specified direction. This operator is trained on a corpus of pick-and-place instructions, such as "move the green object to the right of the red cube," similar to the approach in Kalithasan et al. (2023).

**MCTS Search.** We use an object-centric state representation defined by bounding boxes (including the depth of the center) and visual attributes of all the objects that are present on the table. For each learned inductive concept `<cpt>`, we define a macro-action `Make_<cpt>(size)` that executes the corresponding program with the given size argument, resulting in the construction of the desired concept. Thus, the action space $\mathcal{A}$ is the union of primitive actions $\mathcal{A}_p$ and compound/macro-actions $\mathcal{A}_c$. Intersection over Union (IoU) between the attained state and the expected state in the demonstration is provided as a reward for all macro actions and `keep_at_head(objects)`; all other actions yield zero reward. An MCTS procedure similar to Khandelwal et al. (2016) is performed to find a plan that maximizes the reward. The node expansion process and reward calculation for the MCTS procedure is detailed in Appendix A.3. The search outputs a sequence of grounded actions for an instantiation of the given inductive concept by the task arguments.

**Modularity and Scalability.** MCTS that searches for a plan only in terms of primitive actions may not be generalizable due to lack of modularity C.3. The use of macro-actions in the search ensures that the plan $H_p^*$ for a given demonstration is concise, modular, and easily generalizable. This can be seen as a form of regularization in terms of the length of concept description by making the prior $P(H) \propto |H|^{-\alpha}$ (where $\alpha > 0$) in equation (2)

$$H^* = \arg \min_{H \in \mathcal{H}} \left[ \texttt{Loss}(\{S_1..S_n\}, H(\Lambda)) + \alpha \log |H| \right]$$

However, as the action space expands with the learning of more concepts, the search becomes slower, necessitating the pruning of the search space. To avoid searching over the size parameter in macro-actions, we greedily select the smallest size that achieves the maximum average reward from the current state. Additionally, to prune primitive actions, we train a reactive policy $\pi_{neural}$ which, given the current state $\tilde{s}_t$ and the next expected state $s_{t+1}$ (from the demonstration), outputs one of the primitive actions $a_t^* \in \mathcal{A}_p$. Consequently, the effective branching factor of the search is reduced from $|\mathcal{A}_c| + |\mathcal{A}_p|$ to $|\mathcal{A}_c| + 1$. Thus, our MCTS algorithm is modular through the hierarchical composition of learned concepts and efficient through action space pruning, and is referred to as MCTS+$L$+$P$. Further details regarding modifications in MCTS are given in the Appendix A.3.

## 5.3  (GENERALIZE) PLAN TO PROGRAM ABSTRACTION

Leveraging the code generation and pattern matching abilities of LLMs (Mirchandani et al., 2023), we use GPT-4 to distil out a general Python program from the sequence of grounded actions as determined by MCTS+$L$+$P$. The learnt program is incorporated in the concept library, $\mathcal{L}$, for modular reuse in subsequent learning tasks. Additional details, prompting mechanism and use of learnt programs in the search step of future learning tasks are described in Appendix C.2, A.3. We take a curriculum learning approach beginning from learning of primitive actions and visual attributes,

followed by structures of increasing complexity. Appendix C.2, Fig. 2, A.4 details the curriculum used for concept learning, architecture details, and learning from multiple demonstrations.

# 6 EVALUATION SETUP

**Corpus.** A corpus is created using a simulated Robot Manipulator assembling spatial structures on a table-top viewed by a visual-depth sensor. Demonstration data (3 demonstrations per structure, with up to 20 objects present in the scene) includes observations (via a visual-depth camera) of the action sequence (picking and placing of blocks) resulting in the construction of the final assembly using varied block instances and types (e.g., cubes, dice, lego etc.). The scope of concepts and associated evaluation tasks are adapted from closely related works. The staircase and enclosure construction tasks are inspired from from Silver et al. (2019), adapted to 3D from the original 2D grid world setting. Structures such as boundaries involving repetitive use of columns and rows (w/o explicit joint fastening) are inspired by a robotic assembly data set (Collins et al., 2024). Finally, the arc-bridge and x-shaped patterns are inspired from concept learning works as Lake et al. (2015). A total of 15 structures types are incorporated and are additionally modulated in size/spatial arrangement for generalization evaluation.

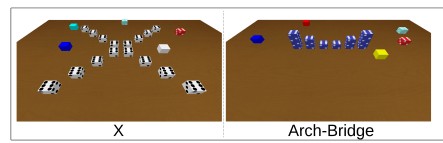

Figure 3: Illustrative examples of spatial structures from corpus showing inductive composition over simpler structures. Details and visualizations in Appendix B

Three evaluation data sets are formed each with *simple* structures and *complex* structures composed of simpler concepts (e.g., staircase consists of towers as substructure). *Dataset I* and *II* contain demonstrations constructing structures with $\texttt{size}(.) \in [3, 5]$, where $\texttt{size}$ is defined in 4. *Dataset II* reverses the linguistic labels used (e.g., the *"tower"* in *I* becomes *"rewot"* in *II*) to assess model reliance on pre-training knowledge in presence of new labels for concepts. *Dataset III* includes concepts of larger size than those in training to test generalization.

**Baselines.** Four baselines are formed from two alternative approaches as follows.

(1) *Purely-Neural*: An end-to-end neural model inspired by StructDiffusion (Liu et al., 2023) that treats structure construction as a rearrangement problem. We consider two variations of the model: (1.1) Struct-Diff (SD): End-to-end approach without any additional supervision regarding which objects need to be moved. (1.2) With-Grounder (SD+G): Similar to (1.1) except that we assume a perfect object selector/grounder that identifies the relevant set of objects which are to be moved.

(2) *Pre-trained models that directly output symbolic programs:* (2.1) LLMs for Scene-Graph Reasoning: This approach uses a Pre-trained Language Model (GPT-4) to generate Python programs from instructions which describe the given demonstration. To help the LLM understand the underlying structure, it is provided with the symbolic spatial relationships (e.g., left(a,b)) between objects in the demonstration. For this baseline we further assume absence of distractor objects in the scene. (2.2) Vision Language Model (GPT-4V): Similar to (2.1) but has the ability to take input demonstration as images. For learning the program of a new inductive concept, we give the demonstration to the VLM in the form of $\Lambda, (S_1..S_g)$. Additional details on prompting method in Appendix C.6. Experiments were also conducted with open source code-generation LLMs such as CodeLlama (70Bq), Due to significantly poorer performances w.r.t. GPT-4, GPT-4 was retained as the primary LLM baseline.

**Model variants.** We implement three variants of the MCTS search to perform a grounded plan search over the action space $\mathcal{A}$: (i) MCTS+*P+L*: Our approach as described in section 5.2, that uses the learnt concepts from $\mathcal{L}$ as macro actions in subsequent searches (e.g., $\texttt{Make\_Tower(3, objects)} \in \mathcal{A}_c$) (*L*). Further it performs pruning of $\mathcal{A}_p$ using $\pi_{\text{neural}}$ (*P*), (ii) MCTS-*P+L*: Our approach without neural pruning and (iii) MCTS+*P-L*: No access to library of concepts during continual learning and thereby lacks ability to use macro actions. This method greedily selects the action from $\mathcal{A}_p$ as given by $\pi_{\text{neural}}$. We provide more details about the 3 methods in Appendix D.4.

**Metrics.** We adopt the following metrics to evaluate our models: (i) *Program Accuracy:* A binary score obtained through human evaluation. 1 for constructing the structure fully, 0 otherwise. (ii) *Target Construction IoU:* Intersection over Union (2D-IoU) between bounding boxes. (iii) *Target Construction Loss:* Mean Squared Error (MSE) loss over the bounding boxes + depth of the center.

# 7 RESULTS

Our experiments evaluate the following questions. **Q1:** How does our model perform when compared to baselines in terms of concept learning and execution ability (In-Distribution)? **Q2:** How does our model generalize to concept instances not seen (larger) during training (Out-of-Distribution)? **Q3:** How robust and efficient is our concept learning pipeline? **Q4:** How can the acquired concepts be used in for embodied instruction following tasks?

**Q1**: CONCEPT LEARNING ACCURACY

We compare the program accuracy (Table 1) and the IoU/MSE values (Table 2) of the final states attained by SPG and the baselines w.r.t. the gold states in the in-distribution setting and find that SPG significantly outperforms other approaches. Values for Purely neural approaches are marked NA because Neural Outputs are not physically grounded. We make the following observations: (i) For *complex* compositional structures, the accuracy of the pre-trained models is poor (zero), indicating their inability to reason over the numerous and complex spatial relations present in these structures. (ii) While program inference via the LLM is better than the VLM for learning *simple* structures, it is worse for *complex* structures. This indicates the inherent weakness of the textual descriptions of complex spatial relations present in complex structures. (iii) While the data-intensive purely neural approaches perform much better on *complex* structures when compared to the pre-trained foundation models, they are still weaker than SPG.

Table 1: Program Accuracy

| Model | Simple | Complex |
|---|---|---|
| SPG(Ours) | **1.00** | **0.83** |
| GPT-4V | 0.33 | 0.00 |
| GPT-4 | 0.78 | 0.00 |
| SD+G | NA | NA |
| SD | NA | NA |

Table 2: In-distribution Performance (Mean $\pm$ Std-error)

| Model | Simple | | Complex | |
|---|---|---|---|---|
| | IoU | MSE (1e-3) | IoU | MSE (1e-3) |
| SPG(Ours) | **0.96** $\pm$ 0.00 | **0.01** $\pm$ 0.00 | **0.85** $\pm$ 0.02 | **2.06** $\pm$ 1.02 |
| GPT-4V | 0.75 $\pm$ 0.01 | 4.33 $\pm$ 0.41 | 0.50 $\pm$ 0.02 | 7.29 $\pm$ 1.10 |
| GPT-4 | 0.89 $\pm$ 0.01 | 1.36 $\pm$ 0.26 | 0.28 $\pm$ 0.02 | 13.5 $\pm$ 1.65 |
| SD+G | 0.74 $\pm$ 0.01 | 1.42 $\pm$ 0.29 | 0.61 $\pm$ 0.02 | 2.43 $\pm$ 0.48 |
| SD | 0.49 $\pm$ 0.01 | 1.48 $\pm$ 0.24 | 0.46 $\pm$ 0.02 | 3.71 $\pm$ 1.53 |

**Q2:** GENERALIZATION PERFORMANCE

Table 3, compares the generalization performance on *Dataset III* for models trained on *Dataset I* (full table in Appendix, 8). We see that SPG outperforms other approaches. We further consider the relative decrease (R.D.) in performance (2D-IoU) on going from the in-distribution to the out-of-distribution (OOD) setting. We make the following observations: (i) SPG suffers a relative decrease of 7.27% for simple and 5.74% for complex structures. (ii) In contrast, the SD+G baseline shows a large R.D. of 63.25% on simple structures and 74.72% on complex structures; highlighting the inability of Purely Neural Models to generalize inductively. (iii) Pre-trained models also have a large R.D. in perf. for complex structures (GPT-4 : 53.87% & GPT4V : 41.64%), which is attributed to their inability to generate the correct program that can generalize inductively to unseen data.

Table 3: OOD Performance. R.D% is the relative decrease in IoU from Table 2. MSE is in 1e-3 units

| Model | Simple | | | Complex | | |
|---|---|---|---|---|---|---|
| | IoU | R.D% | MSE | IoU | R.D% | MSE |
| SPG(Ours) | **0.89** | **7.27** | **0.43** | **0.80** | **5.74** | **1.49** |
| GPT-4V | 0.58 | 23.33 | 13.2 | 0.29 | 41.64 | 10.9 |
| GPT-4 | 0.78 | 12.61 | 5.51 | 0.13 | 53.87 | 19.1 |
| SD+G | 0.27 | 63.25 | 6.21 | 0.15 | 74.72 | 14.2 |
| SD | 0.24 | 51.84 | 6.86 | 0.15 | 67.67 | 11.6 |

Table 4: Perf. on *Dataset II* with Reversed Names. Acc. is Prog. Accuracy, MSE in 1e-3 units

| Model | Simple | | | Complex | | |
|---|---|---|---|---|---|---|
| | Acc. | IoU | MSE | Acc. | IoU | MSE |
| SPG(Ours) | **0.88** | **0.86** | **1.74** | **0.78** | **0.78** | **3.93** |
| GPT-4V | 0.23 | 0.71 | 3.92 | 0.00 | 0.09 | 21.29 |
| GPT-4 | 0.67 | 0.78 | 3.16 | 0.00 | 0.00 | 22.73 |

**Q3:** ROBUSTNESS AND EFFICIENCY ANALYSIS

**Reliance on pre-trained Knowledge vs. Demonstration.** Next, we evaluate the degree to which concept learning relies on prior knowledge vs. the action sequences observed in demonstrations. We

compare pre-trained models against our approach by learning programs on *Dataset II* ( 6), which uses arbitrary names for concepts. This forces all models to rely on demonstration data because there is no real-world knowledge associated with the name of the concept, say *"rewot"* instead of *"tower"*. Table 4 indicates the corresponding performances, with our approach outperforming others. For the IoU/MSE values along with standard errors refer to Appendix Table 9. The relative decrease in performance (program accuracy w.r.t. Table 1) for *simple* structures for our approach (12%) is lower than GPT-4 (14%) and much lower than GPT-4V (30%). The poorer generalization of pre-trained models can be attributed to their over-reliance on prior knowledge and failure to effectively incorporate the data from demonstrations. In contrast, SPG better captures the semantics of a novel concept, especially ones whose knowledge may not be available for the LLMs/VLMs at training time.

**MCTS Variants for Concept Learning.**

Figure 4 compares the program accuracy for the three methods of plan search. For the MCTS-*L+P* method, the program accuracy is expected to be independent of expansion steps as it greedily chooses the action for which $\pi_{neural}$ gives the highest probability. For the MCTS+*L* based methods the accuracy increases beyond 0.6 with time, which demonstrates that having a composable library of concepts allows us to learn a much richer class of inductive concepts. MCTS+*P+L* saturates to a program accuracy of 0.933 in just 4000 expansion steps compared to MCTS+*L-P* taking 512000 expansion steps, which demonstrates a significant increase in learning efficiency via use of the neural pruner. For very low number of expansions steps (<40) accuracy of MCTS+*L* based methods is lower than MCTS-*L* as the former expends expansion steps on UCB exploration (instead of greedy actions).

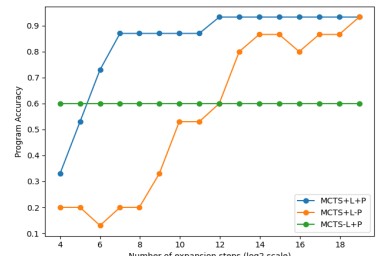

Figure 4: **MCTS Variants.** Num. of expansion steps in search (log scale) (X-axis) vs Program accuracy (Y-axis).

**Significance of MCTS in SPG.** To assess the necessity and importance of MCTS, we carry out an ablation study where we replace it with an LLM planner during the planning stage, referred to as SPG-M+LMP. In the "plan" stage of our pipeline, GPT-4V is prompted to output a plan given the concept library and RGB keyframes from the demonstration. Our experiment shows that GPT-4V struggles to generate correct plans, particularly for complex structures like pyramids, arch_bridge and boundaries, resulting in significantly lower performance than SPG, see 5. Additionally, some plans generated by GPT-4V are not physically grounded, leading to errors in both the planning and generalization stages, which compounds the inaccuracies. This demonstrates that combining symbolic search with LLMs offers a substantial advantage over using only LLMs.

Figure 5: Ablation studies and Disentanglement. Left: Ablations with SPG-M+LMP and GPT-4V+VR. MSE values are in 1e-3. Right: The acquisition of new visual concepts. Plot shows an increase in the likelihood of correct grounding of an object referenced with a new neural concept (*chocolate* color) with training iterations.

| Model | Simple | | | Complex | | |
|---|---|---|---|---|---|---|
| | Acc. | IoU | MSE | Acc. | IoU | MSE |
| SPG(Ours) | **1.0** | **0.96** | **0.01** | **0.83** | **0.85** | **2.06** |
| SPG-M+LMP | 0.55 | 0.68 | 11.1 | 0.16 | 0.19 | 20.0 |
| GPT-4V+VRF | 0.66 | 0.75 | 6.8 | 0.16 | 0.46 | 12.0 |

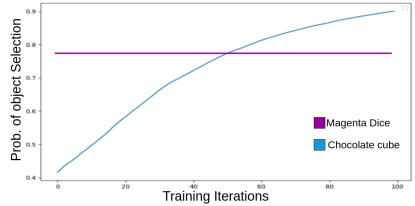

**Effectiveness of Continual Learning of Visual Concepts** Having a disentangled representations allows us to (i) intersperse learning of new visual attributes with learning of new inductive concepts (ii) avoid catastrophic forgetting of already learnt attributes. For example, the model can learn the *chocolate* from an instruction "construct a tower using chocolate blocks of size 4", even if it has not seen the color in the pretraining phase. Because of our modular architecture, we can learn the color as a new embedding in the space of visual attributes. The plot in Fig. 5 demonstrates the benefit of having such disentangled representations. As the training proceeds the probability of being able to select the *chocolate* blocks when required increases with time, while keeping the ability of selecting a magenta colored object (when required) remains the same. Additional details for continual learning of visual concepts appear in Appendix D.2.

**Ablating with Pre-trained Models + Visual Reward Filter** In line with program synthesis techniques using LLMs (Li et al., 2022; Chen et al., 2021), we sample five programs from GPT-4V and rank them according to the visual reward obtained from their execution. Furthermore, we provide the ground-truth programs of the concepts that are needed to learn the given new concept, thus employing a form of teacher forcing in program generation. Even with these measures, it performs significantly worse than SPG, see Table 5 (GPT-4V+VRF). While this performance is better than that of GPT-4V, it still unable to generate correct programs, especially for complex structures.

**Q4:** APPLICATION OF LEARNT CONCEPTS FOR ROBOT INSTRUCTION FOLLOWING

**Complex instruction execution via LLM.** We demonstrate our ability to use the learnt program representations to perform complex guided robot manipulation tasks. We instruct the robot to perform tasks like : "Construct a tower of green die having the same height as the existing tower of white die." and "Construct a tower of total 6 blocks using alternating blue and red blocks". For both the above tasks we prompt GPT-4 by providing it with the set of learnt inductive concepts, the set of primitive actions, and some pre-defined helper functions by using Python import statements in a manner similar to Liang et al. (2023). GPT-4 generates an executable Python code in terms of these functions, which, on running, generates the resultant and required action sequence. Figure 6 (top) illustrates task execution (also see Appendix D.3).

**Grounding learnt concepts into visual input for plan synthesis.** We further demonstrate that the concepts we have acquired can help us to perform goal conditioned planning. Fig. 6 (bottom) demonstrates the results of our approach for the tasks of constructing a staircase beginning from adversarial and assistive initial states. Note, the planner that we learn is a grounded neuro-symbolic planner, as a PDDL based planner cannot be hand-coded easily ( D.5), and LLMs/VLMs are unable to perform such complex reasoning tasks (see Appendix D.5).

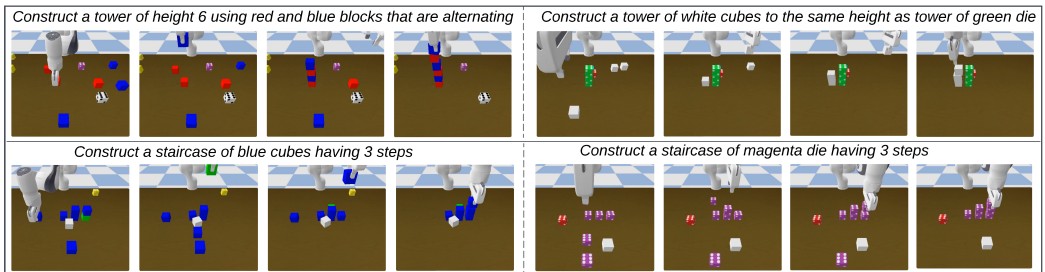

Figure 6: **Application of learnt concepts.** Top: Using LLM to generate the executable code for a novel tasks, given the concept definitions. Bottom: Integrating a neuro-symbolic planner over the concepts. Bottom-left: The planner is able to optimally replace the green cube from the adversarial initial state by unstacking and re-stacking the faulty tower. Bottom-right: The planner is able to complete a staircase from an initially constructed row by layering rows upon rows, a method of construction it has not seen while learning staircase.

## 8   CONCLUSION

This paper introduces a novel approach for learning inductive representation of grounded spatial concepts as neuro-symbolic *programs* via language-guided demonstrations. Our approach factors program learning as: *Sketch:* generating the high-level program signature via an LLM, *Plan:* searching for a grounded plan that maximises the total discounted reward with the respect to the demonstration, and *Generalize:* abstracting the grounded plan into an inductively generalize-able abstract plan via an LLM. Continual learning is achieved via learning of modular programs by giving preference to shorter programs through composition of learnt ones. Extensive evaluation demonstrates accurate program learning and stronger generalization in relation to purely LLM based as well as purely neural baselines. Grounding of learned concepts in visual data facilitates reasoning and planning for embodied instruction following. Limitations include reliance on perfect demonstrations, assumption of full observability of all objects and experiments confined to simulation. Incorporating noisy demonstrations, reasoning with beliefs and interleaving planning and execution remains part of future work.

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

# A   ADDITIONAL DETAILS ON TECHNICAL APPROACH

Figure 7 illustrates the pipeline for online inference to realize to realize construction of novel structures.

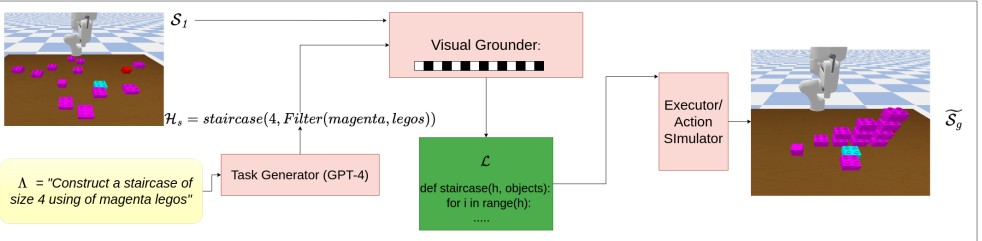

Figure 7: **SPG: Inference** First the library of concept $\mathcal{L}$ is loaded with the corresponding set of learnt programs. Then the given instruction is converted into task-sketch $H_s$, which is grounded in the initial scene. The required program is fetched from the library, and the grounded task-sketch is executed based on the semantics of the learnt program.

## A.1   SYMBOLIC CONSTRUCTS AND THEIR SEMANTICS USED IN PROGRAMS

Table 5 defines the types of the signature and semantics of all the operators. Table 6, includes the type definition of various symbols. Standard Python constructs such as for loops, if else $\cdots$) as assumed in addition to the constructs defined here.

Table 5: **Symbols and Semantics** Signature and semantics for the primitive concepts and operations that are used in the construction of the programs used to express inductive spatial concepts.

| Function | Signature | Semantics |
|---|---|---|
| filter | (VisualConcept, ObjSet) → ObjSet | Returns the objects that contain the VisualConcept |
| move_head | (Head, Dir) → Head | Moves the head to the given direction (May or may not take input/return the head, based on a flag) |
| assign_head   a.k.a move_head(overloaded) | (Head, ObjIdx) → Head | Given the index/one-hot representation for an object, it moves the head to the position corresponding to that object |
| keep_at_head | (ObjSet, Head) → None | Keeps the argmax of ObjSet at the head |
| reset_head | None → Head | Sets the head to the top position of stack and pops this position from the stack as well |
| store_head | Head → None | Pushes the current position of head into the stack |

## A.2   CURRICULUM LEARNING

We follow a curriculum approach where the visual concepts are trained first from simpler linguistically-described demonstrations ($t_0$ in figure 8). This is followed by learning of action concepts through sequentially composed pick and place tasks. Its essential to use such long range sequential instructions in order to ensure that the semantics of action concepts are learnt for placement of objects at a height much above the tabletop ($t_1$ in figure 8). After the pre-training phase, the agent can continually learn new inductive concepts and visual attributes. ($t_2, t_3$ in figure 8)

Table 6: **Symbolic representation.** The table lists the type definitions used in the implementation of SPG programs.

| Defined Types | Python Type | Usage |
|---|---|---|
| IntArg | int | Argument for the structures that defines the size (height, length etc) |
| Obj | torch.Tensor | One-hot vector whose non-zero index represents the selected objects |
| ObjSet | torch.tensor | Probability mask over the selected objects |
| Dir | string | Primitive directions like left, right, front, top, etc |
| ConceptName | string | name of the visual, action or inductive concept |
| Head | torch.Tensor | Bounding box with depth. 3D cuboidal space. |

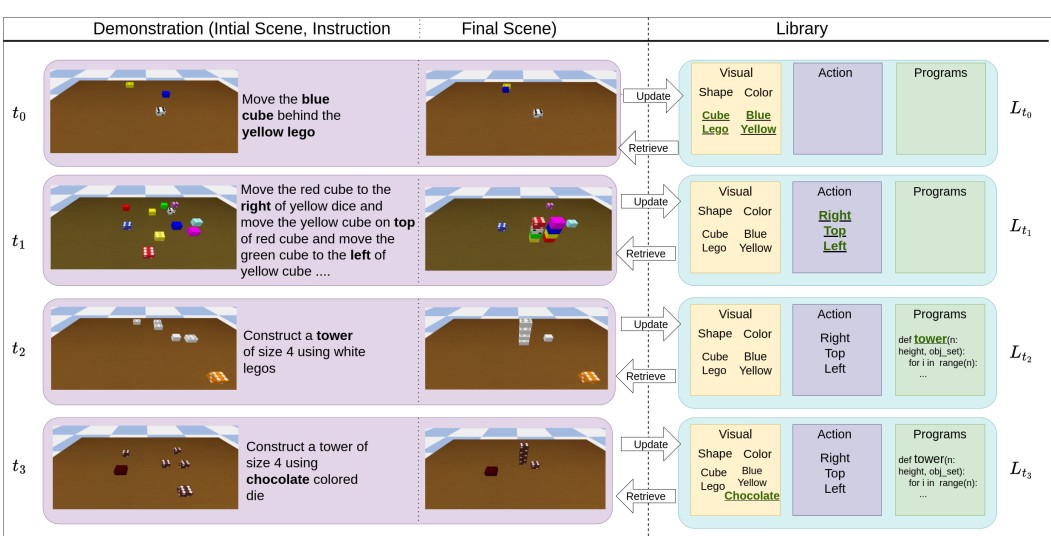

Figure 8: **Continual learning through curriculum:** Using simple pick and place demonstrations we learn visual attributes such as *blue cube, yellow lego* ($t_0$). Using long range instructions which are sequentially concatenated descriptions of pick and place tasks we train our action concepts such as *left, right, top* ($t_1$). After pre-training the agent can perform continual learning of concepts such as learning generalized representation for *tower* ($t_2$). Because of disentangled representation of neural and symbolic concepts, interspersed learning of new visual attributes such as *chocolate color* are also possible through few demonstrations of structure creation($t_3$).

## A.3  DETAILS FOR PLAN-SEARCH AND GENERALIZATION

**Modifications to the Simulation and Reward Back propagation Steps**: Next, we outline the modifications in the simulation and the reward back propagation steps of the standard MCTS algorithm for our setting. During program search we assume access of intermediate scenes in the demonstration. This allows us to provide intermediate rewards that can guide the search well. We observed that making the following changes in simulation and back propagation step increased the efficiency of our search procedure. Fig 9 illustrates the possible states explored by MCTS and the reward calculation.

- Simulation: Rather than performing Monte Carlo simulations at each newly expanded leaf node (to estimate its value) we completely avoid these simulation steps. This was motivated by the fact that our reward is not completely sparse and the intermediate IoU rewards for each object we place allow us to guide the search effectively.

- Back propagation: We perform off policy Q-learning updates during back propagation similar to one indicated by Khandelwal et al. (2016) :

$$V(s_t) = max_{a \in \mathcal{A}} Q(s_t, a). \tag{4}$$

$$\tau(s_t, a) = s_{t+1} \tag{5}$$

$$Q(s_t, a) = r_t + \gamma V(s_{t+1}) \tag{6}$$

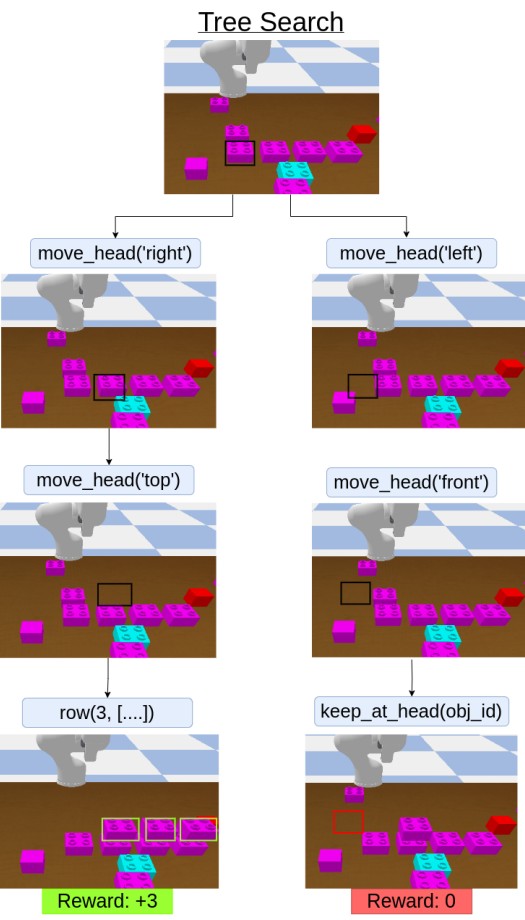

Figure 9: A sample MCTS search tree outlining the states explored and the calculation of reward.

**Improving Modularity and Scalability of MCTS procedure:** We present additional details on the MCTS procedure for searching for a plan conditioned on a program signature and guided by the demonstration.

In order to incorporate the objective of searching for physically realizeable plans and facilitating generation via re-use of concepts, the following conceptual changes are incorporated in the standard MCTS procedure Sutton & Barto (2018).

**Modularity (MCTS + L):** We want to allow learning of novel inductive concepts in terms of existing ones. This would ensure that the plan $H_P^*$ corresponding to a given demonstration is concise and can be easily generalized to the $H_G^*$. C.3 in appendix give example of two plans for the structure *Pyramid* one which is modular and can be successfully generalized by GPT-4, other for which GPT-4 fails in generalization due to lack of modularity. This can be seen as a form of regularization in terms of the length of concept description, by making the prior $P(H) \propto |H|^{-\alpha}$ (where $\alpha > 0$) in equation (2)

$$H^* = \arg \min_{H \in \mathcal{H}} [\text{Loss}(\{S_1..S_g\}, H(\Lambda)) + \alpha \log |H|] \tag{7}$$

In order to allow modular learning of programs, for every inductive concept already stored in the library we define corresponding action instantiations which can be a potential candidate actions during our search. As an example, for the concept `Tower` we have one of the action instantiation as `Make_Tower(3, objects)` which would be the action of constructing desired tower. This can be visualized as a compound/macro-action which is composed of primitive actions `keep_at_head(objects)`, `move_head('top')`. We define $\mathcal{A}_c$ as the space of such compound actions, and $\mathcal{A}_p$ as the space of primitive action/function consisting `reset_head()`, `move_head(direction)`, `keep_at_head(objects)`, `store_head()`. Realization of equation (6) (increased preference of macro-actions over primitive ones) is done through discounted IoU rewards during our search (`Make_Tower(3)` would have a reward of 1+1+1, as compared to $1 + \gamma^2(1) + \gamma^4(1)$ for a sequence of 3 (`keep_at_head(objects)`, `move_head('top')0`)). We refer to MCTS using concept instantiations from $\mathcal{L}$ as macro actions in search as MCTS+*L*.

**Scalablility (MCTS+*P*)**: as more concepts are added to the library $\mathcal{L}$ the action space of our search $\mathcal{A}$ (specifically $\mathcal{A}_c$, the space of compound concepts) increases, therefore we want to prune the search space effectively. For this during the pre-training phase we train a reactive policy $\pi_{neural}$ which given the current state, $\tilde{s}_t$ and the next expected state $s_{t+1}$ (part of the demonstration) would output one of the primitive action, $a_t^* \in \mathcal{A}_p$ where $\mathcal{A}_p$ is the primitive action space, i.e. $a_t^* = \pi_{neural}(a_t|\tilde{s}_t, s_{t+1})$, $a_t \in \mathcal{A}_p$ Note that while expanding our search tree we only search among the space of compound actions $\mathcal{A}_c$ and the action $a_t^*$, thereby reducing the branching factor of search from $|\mathcal{A}_c \cup \mathcal{A}_p|$ to $|\mathcal{A}_c| + 1$. We refer to MCTS using neural pruning as MCTS+*P*. Therefore our MCTS algorithm is modular through hierarchical composition of learnt concepts and efficient through pruning of action space and is referred to as MCTS+*L*+*P*.

**Generalization**: GIven multiple equal length plans for a given demonstration, we seek to recover a plan one that can be easily generalized by the LLM. 5 shows a plan which could be correctly abstracted out into a generic program by GPT-4. Whereas 15 shows another plan with similar semantics, for which GPT-4 is unable to correctly find the generalized program (Note that row or column of size 1 is equivalent to keep_at_head). We tackle this problem in the following manner.

1. Rather than getting a single plan from the plan search we get the top k plans $\{H_{P,i}\}_{i=1}^{i=k}$. In order to get these top k plans we expand the complete tree (based on UCB criteria) starting from the root node corresponding to the initial state, till a predefined budget of expansions. Then we select the top k paths(potential plans) from the root node to all the leaf nodes (where the top k ones are those that give the highest accumulated IoU reward with respect to the given demonstration).

2. Later we abstract out each of these k plans into corresponding generalized programs, $\{H_{G,i}\}_{i=1}^{i=k}$ using GPT-4. We again run each of these programs on the given demonstration and then choose the one which gives the highest IoU reward (resolving ties based on predefined order). Note that some program $H_{G,i}$ upon execution may result in a plan $\tilde{H}_{P,i}$ different from the original plan $H_{P,i}$ using which it was generalized. This can be attributed to potential errors in GPT-4s program generalization process.

## A.4 Additional Details: Learning with Increasing Number of Demonstrations

Given *k* demonstrations for a novel inductive concept, we independently find k task sketch $\{H_{S,i}^*\}_{i=1}^{i=k}$ and grounded plans $\{H_{P,i}^*\}_{i=1}^{i=k}$. During the generalization phase we give these k pair of task-sketch and corresponding plans to GPT-4 and ask into infer a single abstraction over them. C.5 in appendix gives a concrete example. Equation for generalize step (getting $H_G^*$ from $H_P^*, H_S^*$) can be modified as follows.

$$H_G^* \leftarrow Generalize(H_G \,|\, \{H_{P,i}^*, H_{S,i}^*\}_{i=1}^{i=k}; \theta_G), \quad H_G \in \mathcal{H}^\mathcal{G} \tag{8}$$

## A.5 Detailed Experimental Methodology

**Input to the Method:** The input consists of a language instruction and a human demonstration represented as a sequence of RGBD keyframes.

**Output/Aim of the Model:** The goal is to learn a representation of the unknown concept in the instruction, assuming there is only one unknown concept. If the unknown concept is inductive (e.g.,

"tower"), the model learns a program definition $def\,tower()$ and stores it in the program library. If the unknown concept is a primitive concept, a concept embedding is learned through backpropagation.

**Evaluation:** The learned model is evaluated based on the correctness of the program representation for inductive concepts and the correctness of object placements, measured through the Intersection over Union (IoU) metric (see Metrics, line 262). d. Examples: Suppose the current library contains the concepts "red", "tower". Given the instruction "construct a staircase of height 3 using red blocks," the process is as follows:

**Parsing:** The instruction is parsed into a sketch: Staircase(height=3, objects=filter(red, blocks)).

**Grounding:** The "objects" parameter is grounded using the visual grounder, which identifies the indices of the red blocks, e.g., [1, 2, 3, 4, 5, 6]. That is, filter(red, blocks) = [1,2,3,4,5,6].

**Planning:** The planning step uses the demonstrations (sequence of keyframes) to identify the sequence of actions that best explains the demonstrations. In this case, the plan might be: Tower(height=1, objects=[1,2,3,4,5,6]), move_head(right), Tower(height=2, objects=[2,3,4,5,6]), move_head(right), Tower(height=3, objects=[4,5,6]

**Generalization:** The generalization step abstracts the plan obtained from three such demonstrations into a program. The resulting program would be:

```
1 def staircase(height, objects):
2     for i in range(height):
3         tower(height=i, objects)
4         move_head(right)
```

*Note:* Whenever an object is placed, the objects list is modified in place, and the index of the placed object is removed. primitive actions: The movement of any object is achieved by first determining the placement position by moving the head (an imaginary bounding box) in specific directions and then placing the object to be moved at the head. The head is implemented as a 3D bounding box defined by coordinates (x1, y1, x2, y2, d), where x1, y1, x2, and y2 are the 2D corners of the bounding box, and d is the depth at the center. The primitive action move_head(direction) shifts the bounding box in the required direction. The action keep_at_head(object_list) picks the first object in the list and places it at the center of the bounding box. Two other primitives, store_head and reset_head, are used to save the current position of the head, allowing the search to return to useful positions later if needed.

## B  ADDITIONAL DETAILS REGARDING DATASETS

Figure 10 demonstrates the kind of inductive concepts for which we want to learn generic (i.e instance agnostic) representations.

**Dataset for Pre-training:** We use 5k examples of constructing *twin-towers* i.e. 2 towers adjacent to each other, for learning semantics of move_head(dir), a basic set of visual attributes, reactive policy $\pi_{neural}$, and neural modules required for grounded planning. The twin towers allow us to learn various action semantics for all possible configurations of blocks in 3D-space (and not being limited to blocks placed directly on table top surface). Since we are not aware of the underlying semantics of *tower* during pre-training phase the corresponding natural language instruction consists of step by step pick and place actions. 11 gives example demonstrations from this dataset.

**Dataset for Inductive Structures:** We learn a variety of structures which we have divided into *Simple* and *Complex* structures. A structure is considered complex if it can be expressed as an inductive composition of simpler structures. As an example, we can express a staircase to be a composition of towers of increasing height. The structures are listed in the Table 7. Figure 12 shows the hierarchical relationship among these structures in the form of a DAG (directed acyclic graph). F.3 gives the ground truth program representations for each structure.

## C  PROMPTING STRATEGY AND EXAMPLES

This section gives various prompting examples for our approach and baselines, along with examples motivating particular design decisions in our approach.

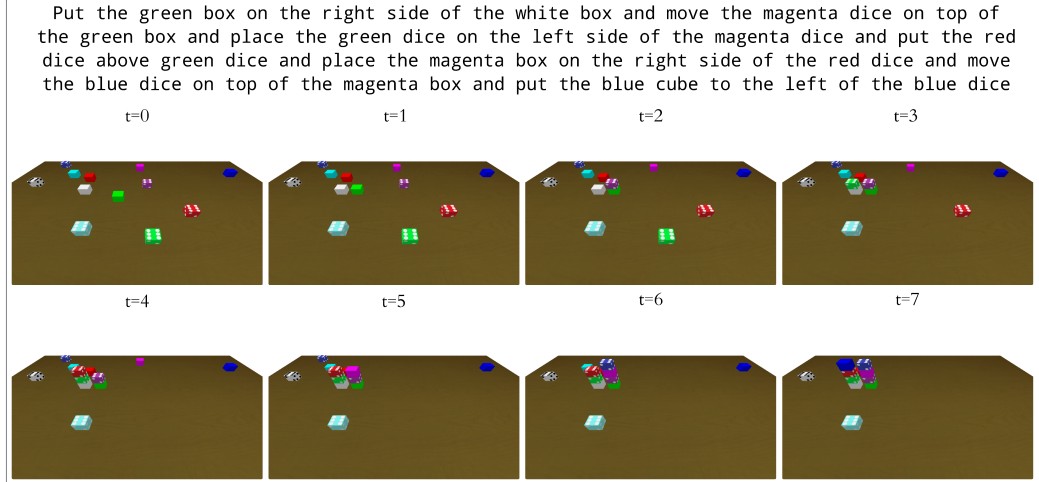

Figure 10: **Illustration of the inductive concepts.**

Put the green box on the right side of the white box and move the magenta dice on top of the green box and place the green dice on the left side of the magenta dice and put the red dice above green dice and place the magenta box on the right side of the red dice and move the blue dice on top of the magenta box and put the blue cube to the left of the blue dice

Figure 11: **Example from pre-training dataset.**

Table 7: **Structure Types.** Examples of simple and complex structures considered in this work for the robot to construct.

| Simple Structures | Complex Structures |
|---|---|
| Row, Column, Tower
Inverted-Row, Inverted-Column
Diagonal-45, Diagonal-135
Diagonal-225, Diagonal-315 | X (cross-shape), Staircase
Inverted-Staircase, Pyramid
Arch-Bridge, Boundary |

## C.1    PROMPT EXAMPLE FOR TASK SKETCH GENERATION STAGE (*Sketch*)

In order to get a program representation (high level task sketch) of the given natural language instruction, we prompt GPT-4 with few shot examples in a manner similar to Liang et al. (2023). Code segment 1 gives an example of getting the task sketch given the demonstration for constructing a *staircase*. We first import the available primitive operators and functions and also give examples in order to demonstrate the signature of the available primitives(line 1-8). Then we give incontext

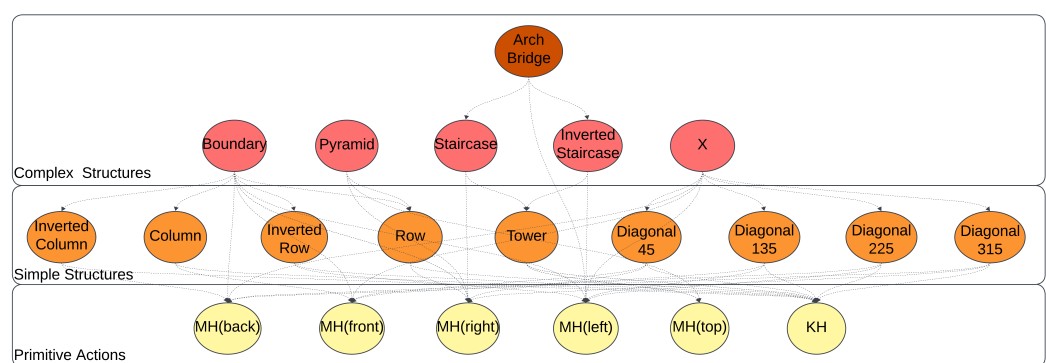

Figure 12: **Hierarchy of the structures/programs.** This diagram shows the hiearchical nature of the structures in our dataset. MH is abbrevation for move_head and KH is abbreviation for keep_at_head

examples of how to parse various natural language instructions in a program representation(line 10-14). We append to this prompt the instruction for current task(line 16-17).

```
1  # importing the available functions
2  from visual_operators import filter
3  from inductive_operators import get_parameters, find_structure
4
5  # function signature of the imported functions
6  filter(color, cube) # filter the objects that are cubes and color ..
7  get_parameters(structure) # parameters of the structure ...
8  find_structure(type, description) # finding structure of given type,
       description
9
10 # examples:
11 # instruction: Find the tower with green cubes
12 find_structure(type = tower, description = filter(green, cube))
13 # instruction: Construct a tower of height 3 with yellow cubes
14 Tower(height = 3, objects = filter(yellow, cubes))
15
16 # current task: Construct a staircase of 4 steps using cyan legos
17
18 # (GPT-4s output)
19 Staircase(steps = 4, filter(cyan, legos))
```

<p align="center">Listing 1: Task Sketch Generation Using GPT-4 (Sketch)</p>

### C.2 PROMPT EXAMPLE FOR GENERALIZING A SEQUENCE OF ACTIONS/PLAN TO A GENERAL PROGRAM (*Generalize*)

Code segment 2 give an example of getting the general Python program from the plan found using MCTS. We first provide a base prompt giving details to GPT-4 about the desired task (line 1-2). Then we give the input arguments and the corresponding output/plan for a given demonstration (line 3-5). We expect the GPT-4 to output the final Python program (line 7-10)*staircase* (line 6-8).

```
1  # Write a general python code which on the given input produces the
       desired output, do not output anything other than the function
       description.
2
3  # input: n = 4, objects = ObjSet
4  # output: tower(1, ObjSet), move_head('right'), tower(2, ObjSet),
       move_head('right'), tower(3, ObjSet), move_head('right'), tower(4,
       ObjSet)
5
6  # Program (GPT-4s output)
7  def staircase(n, objects):
```

```
8       for i in range(n):
9           tower(i+1, objects)
10          move_head('right')
```
Listing 2: Plan to Program using GPT-4 (Generalize)

## C.3 BENEFIT OF ESTIMATING MODULAR/SMALLER PLANS

The below examples demonstrate the benefit of learning new inductive concepts in terms of already acquired inductive concepts (more modular representation). The program 3 is obtained through generalization of plan $H_p$ that represents *pyramid* in terms of rows of decreasing size. The generated program generalizes to *pyramid* of different height.

```
1  # input: n = 3, objects = ObjSet
2  # output: row(5, ObjSet), move_head('right'), move_head('top'), row(3,
       ObjSet), move_head('right'), move_head('top'), row(1, ObjSet)
3
4  # Program (GPT-4s output)
5  def pyramid(n, objects)
6   for i in range(n, 0, -1):
7           # Calculate the number of objects in the current row
8           row_count = 2 * i - 1
9
10          # Perform actions for the current layer
11          row(row_count, objects)
12          if i > 1:  # No need to move head after the last row
13              move_head('right')
14              move_head('top')
```
Listing 3: Pyramid in terms of rows

On the other hand program 4 which is obtained in an attempt to generalize the plan expressed in terms of primitives is incorrect.

```
1  # input: n = 3, objects = ObjSet
2  # output: store_head(), keep_at_head(ObjSet), move_head('right'),
       keep_at_head(ObjSet), move_head('right'), keep_at_head(ObjSet),
       move_head('right'), keep_at_head(ObjSet), move_head('right'),
       keep_at_head(ObjSet), reset_head(), move_head('top'), move_head('
       right'), store_head(), keep_at_head(ObjSet), move_head('right'),
       keep_at_head(ObjSet), move_head('right'), keep_at_head(ObjSet),
       reset_head(), move_head('right'), move_head('top'), keep_at_head(
       ObjSet)
3
4  # Program (GPT-4s output)
5  def pyramid(n, objects)
6  # Store initial position
7       store_head()
8
9       # Initial sequence of placing and moving right
10      for _ in range(n):
11          keep_at_head(objects)
12          move_head('right')
13
14      # Reset and move to top right position
15      reset_head()
16      move_head('top')
17      move_head('right')
18
19      # Store again and repeat placing and moving right
20      store_head()
21      for _ in range(n):
22          keep_at_head(objects)
23          move_head('right')
24
```

```
25      # Reset and adjust final position
26      reset_head()
27      move_head('right')
28      move_head('top')
29      keep_at_head(objects)
```

Listing 4: Pyramid in terms of primitive actions

### C.4 COMPARING SAME-LENGTH PLANS IN TERMS OF GENERALIZABILITY

Below we illustrate plan to program generalization for 2 different plans which are correct and have same length, but LLMs find it difficult to generalize the second one as compared to first. (Note that row and column of size 1 are equivalent to keep_at_head)

```
1 # input: n = 3, objects = ObjSet
2 # output: keep_at_head(ObjSet), move_head('top'), keep_at_head(ObjSet),
    move_head('top'),  keep_at_head(ObjSet)
3
4
5 # Program
6 def tower(n, objects):
7     for _ in range(n):
8             keep_at_head(objects)
9             move_head('top')
```

Listing 5: Plan for tower that can be easily generalized (correct generalization)

```
1 # input: n = 3, objects = ObjSet
2 # output: row(1, ObjSet), move_head('top'), keep_at_head(1, ObjSet),
    move_head('top'),  column(1, ObjSet)
3
4
5 # Program
6 def tower(n, objects):
7     for i in range(1, n + 1):
8             row(i, objects)
9             move_head('top')
10            keep_at_head(objects)
11            move_head('top')
12            if i < n:
13                column(i, objects)
14                move_head('top')
```

Listing 6: Plan for tower that is difficult to generalize (Incorrect generalization)

### C.5 GENERALIZING VIA MULTIPLE DEMONSTRATIONS

Given multiple demonstrations we independently find task sketch and corresponding grounded plans for each demonstration. These are further given to GPT-4 for generalization. Code segment 7 gives an example of getting a single Python program from multiple demonstrations. Note that we explicitly prompt the LLM that some of the grounded plans might be incorrect (which may lead to more robust generalization in case of noisy demonstrations).

```
1
2 # Function Call: wor(height = 3, objects = ObjSet_1)
3 # Execution: keep_at_head(obj = ObjSet_1), move_head(dir = right),
    keep_at_head(obj = ObjSet_1), move_head(dir = right), keep_at_head(
    obj = ObjSet_1),
4 # Function Call: wor(height = 3, objects = ObjSet_1)
5 # Execution: keep_at_head(obj = ObjSet_1), move_head(dir = right),
    keep_at_head(obj = ObjSet_1), move_head(dir = right), keep_at_head(
    obj = ObjSet_1),
6 # Function Call: wor(height = 3, objects = ObjSet_1)
```

```
 7  # Execution: column(size=1, obj = ObjSet_1), move_head(dir = right),
        keep_at_head(obj = ObjSet_1), move_head(dir = right), keep_at_head(
        obj = ObjSet_1),
 8
 9  #Write the function definition, which generalizes the above executions.
        Note that some of the executions can be partially wrong.
10  ```python
11  def wor(height, objects):
12  ```
13
14  GPT-4s Output .....
```

Listing 7: Generalizing through multiple plans

## C.6 PROMPT EXAMPLES FOR LEARNING PROGRAMS USING LLM/VLM MODELS

Below we describe the prompting methodologies for learning programs through LLM/VLM models. Note that although the prompt examples described below are for the case of learning novel structure from 1 demonstration, we use 3 demonstration per novel structure in our main results (for both our approach and LLM/VLM baseline).

**LLM/GPT-4** Code segment 8 depicts our prompting methodology given a demonstration for a new concept *tower*. For this baseline we aim to check demonstration following and spatial reasoning abilities of LLMs (GPT-4). We provide supervision of the intermediate scenes by using tokenized spatial relations between objects in the scene (Given in the form of Scene = [right(1, 0) ...]). We further assume that only those objects that are required to perform the task are present in the scene (no distractor objects). For every structure (that needs to be learned at time t) we give LLM a prompt providing in-context example on how to generalize (line 19-35), the set of primitive operators (line 4) available and the set of structures learnt/present in library (till time t-1) (line 5-18). Finally we append to this prompt the expected declaration (arguments and keywords arguments) of the inductive concept that is to be learnt along with the spatial relations for each scene of the given demonstration (36-51). Note that we assume absence of distractor objects for this baseline.

```
 1  # Consider a block world domain ...... Given a structure creation task
        along with intermediate scnes complete a general Python function for
        it. The function should be in terms of primitive operators and
        already learnt structures that are present in the program library.
        Enclose the function within backtick (```)
 2
 3  primitive_operators = [keep_at_head, move_head ..]
 4  # this would be our program library
 5  learnt_structures = {
 6      "row": {
 7              "program_tree":
 8              '''
 9              def row(size, objects):
10                  for i in range(size):
11                      keep_at_head(obj = objects)
12                      move_head(dir = 'right')
13              ''',
14          },
15
16      ......
17  }
18  # the example task
19  Example task:- Place all the objects to the right of each other.
20  Final state :- [right(1, 0), right(2, 1), right(3, 2), right(4, 3)]
21  Intermediate scenes :-
22  Scene 0 = []
23  Scene 1 = []
24  Scene 2 = [right(1, 0)]
25  Scene 3 = [right(1, 0), right(2, 1)]
26  Scene 4 = [right(1, 0), right(2, 1), right(3, 2)]
27  Scene 5 = [right(1, 0), right(2, 1), right(3, 2), right(4, 3)]
```

```python
28 Python function :-
29 ```python
30 def placing_all_right(objects):
31     for i in range(len(objects)):
32         keep_at_head(objects) # select one object from the objects set
       and keep the head at this location
33         move_head(dir = 'right') # move the head to the right of the
       previous position
34 ```
35 # The current task for which program needs to be found
36 Current task:- Construct a tower of size 6.
37 Final state :- [top(1, 0), top(2, 1), top(3, 2), top(4, 3), top(5, 4)]
38 Intermediate scenes :-
39 Scene 0 = []
40 Scene 1 = []
41 Scene 2 = [top(1, 0)]
42 Scene 3 = [top(1, 0), top(2, 1)]
43 Scene 4 = [top(1, 0), top(2, 1), top(3, 2)]
44 Scene 5 = [top(1, 0), top(2, 1), top(3, 2), top(4, 3)]
45 Scene 6 = [top(1, 0), top(2, 1), top(3, 2), top(4, 3), top(5, 4)]
46 Python function :-
47 ```python
48 def tower(size, objects):
49 ??
50 ```
```

Listing 8: Prompting Strategy for LLM baselines (GPT-4)

**VLM/GPT-4-V** Unlike LLM, VLMs have the abilities to process the demonstration as a sequence of visual frames. Therefore rather than providing the symbolic spatial relations between every scene we instead directly provide all the intermediate scenes for the given demonstration. Further we also relax the assumption that there are no distractor objects. As shown in figure 13 We first give information about the set of primitive operators and the structures that we have already learnt (library of concepts). In order to visually ground the semantics of our primitive actions we give 3 example tasks (natural language instruction and intermediate scenes) that do not directly refer to any structure, along with corresponding sequence of actions taken (*# Demonstration for visual grounding*). We further provide another example (without scenes) demonstrating how to write generalizable Python function for a given task using our operators (*# Example for generalization*). Finally we give the natural language instruction and corresponding scenes for the current task along with signature of the program to be learnt (*# Current task description*).

Figure 13: **Prompt example for VLM-baseline.** Figure shows the prompting strategy for GPT-4-V that includes the primitive actions, the example tasks for visual grounding, example of writing generalizable Python functions and the demonstration frames.

# D    SUPPLEMENTARY RESULTS

**Out-of-Distribution Performance:**

Table 8: Out-of-Distribution Performance (mean $\pm$ std-error)

| Model | Simple | | Complex | |
|---|---|---|---|---|
| | IoU | MSE | IoU | MSE |
| SPG(Ours) | **0.892** $\pm$ 0.065 | **4.386e-4** $\pm$ 2.387e-4 | **0.804** $\pm$ 0.025 | **0.001** $\pm$ 5.391e-4 |
| GPT-4 | 0.776 $\pm$ 0.023 | 0.006 $\pm$ 0.001 | 0.131 $\pm$ 0.019 | 0.019 $\pm$ 1.498e-3 |
| GPT-4V | 0.575 $\pm$ 0.026 | 0.013 $\pm$ 0.001 | 0.290 $\pm$ 0.016 | 0.011 $\pm$ 1.316e-3 |
| SD | 0.236 $\pm$ 0.005 | 0.006 $\pm$ 7.495e-4 | 0.150 $\pm$ 0.011 | 0.011 $\pm$ 2.860e-3 |
| SD+G | 0.273 $\pm$ 0.004 | 0.006 $\pm$ 6.180e-4 | 0.154 $\pm$ 0.010 | 0.014 $\pm$ 2.958e-3 |

**Performance *Dataset II* (i.e. name reversed evaluation):** Table 4 gives the corresponding program accuracies, while Table 9 give the corresponding IoU/MSE metrics along with standard errors.

## D.1    QUALITATIVE COMPARISON BETWEEN PURELY-NEURAL (STRUCT-DIFF+GROUNDER) VS. OURS(SPG)

Figure 14 gives compares the qualitative results for our approach against Struct-Diffusion with grounder on both in-distribution, *Dataset I* and out-of-distribution (larger size), *Dataset III*. In in-distribution setting the our method performs slightly better in terms of structure creation for both simple and complex structures, but the difference is not significant. However for out-of-distribution setting structures created by our approach are much better than those created by Struct-

Table 9: Performance on *Dataset II*(Names Reversed)

| Model | Simple | | Complex | |
|---|---|---|---|---|
| | IoU | MSE (1e-3) | IoU | MSE (1e-3) |
| SPG(Ours) | **0.86** ± 0.03 | **1.74** ± 0.45 | **0.78** ± 0.02 | **3.93** ± 1.09 |
| GPT-4 | 0.78 ± 0.03 | 3.16 ± 0.51 | 0.00 ± 0.00 | 22.73 ± 1.48 |
| GPT-4V | 0.71 ± 0.01 | 3.92 ± 0.45 | 0.09 ± 0.02 | 21.29 ± 1.59 |

Diffusion+Grounder. Further for this setting structure creation by Struct-Diffusion seems to be much worse for complex structures than simple ones.

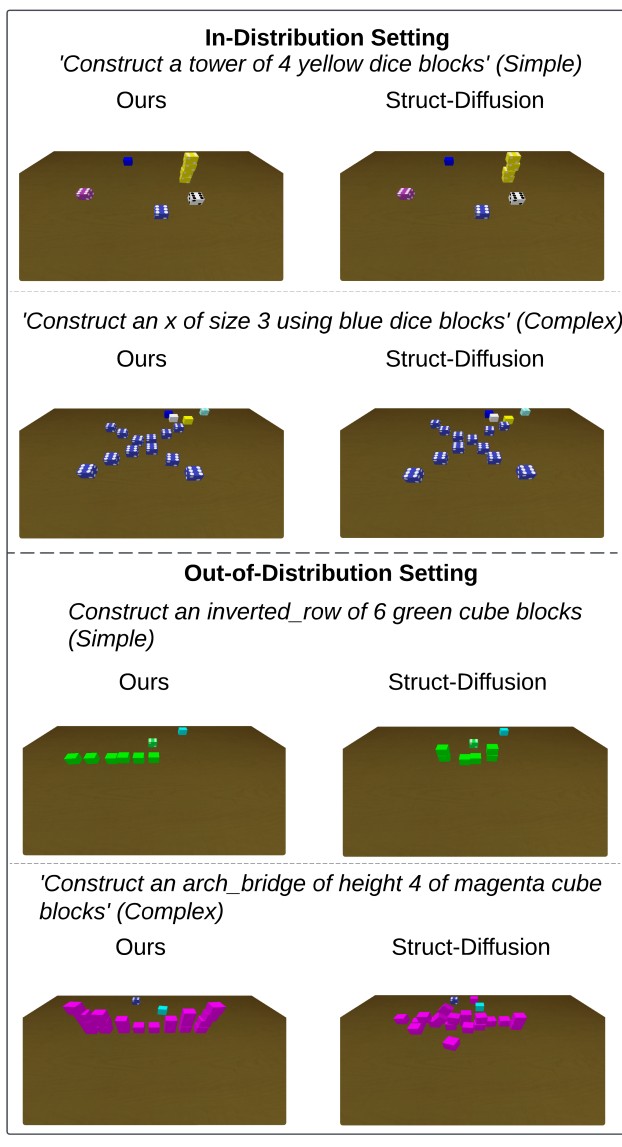

Figure 14: **Structure creation comparison between SPG(Ours), and Struct-Diff+Grounder**

## D.2 CONTINUAL LEARNING OF NEURAL CONCEPTS

Given demonstration for the task "Construct a tower of height 4 using *chocolate* cubes", we would like to learn the neural embedding for the unknown color *chocolate* (where we assume that tower has

already been learnt and stored in the library $\mathcal{L}$). First the instruction is converted into corresponding plan sketch $\mathcal{H}_S$ = tower(4, Filter(**chocolate**, cubes)), which is passed to the visual grounder. The grounder detects the presence of an unknown attribute *chocolate* as an argument to filter, and randomly initializes a new neural embedding for it. Using this new embedding along with the already present embedding of cube and the visual features found through ResNet-34, the quasi-symbolic executor outputs a grounded task-sketch. The executor executes the grounded task-sketch by getting the semantics of the underlying function i.e. tower from the library $\mathcal{L}$. MSE+IoU loss computed over the final scene obtained and the expected final scene is backpropagated through the network. Note that during backpropagation all the neural modules (action semantics, visual attributes, ResNet-34) are frozen, except for the newly initialized embedding for *chocolate*. For the purpose of differentiable sampling during tower construction we use gumbel-softmax Jang et al. (2017) with masking. Figure 15 illustrates our approach.

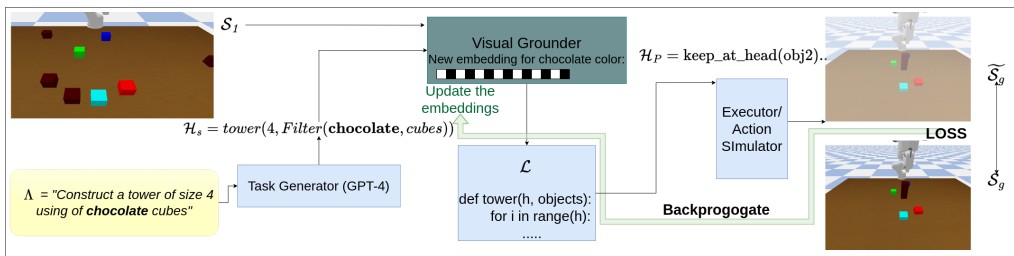

Figure 15: **Continual learning of visual primitives**

### D.3 DETAILS FOR INFERENCE ON NOVEL TASKS USING AN LLM

Below we show the Liang et al. (2023) inspired prompting methodology that we use to get the executable code corresponding to a language specified manipulation task. We initially begin by importing the helper functions, spatial direction, primitive functions, and learnt inductive concepts/structures (line 5-11). Then we give few examples for how to use and compose the various functions for different tasks (line 16-83). Finally we give the instruction of current task, and expect GPT-4 to output the corresponding executable code.

```python
1  # Given a task you have to provide Python code for executing the task
2
3  # importing available functions
4
5  from spatial_directions import top, front, back, left, right
6
7  from primitives import assign_head, move_head, keep_at_head
8  # HEAD is a imaginary pointer keeping track of the current spatial
        location in consideration
9
10 from helpers import find_size, filter
11 from structures import row, column, tower
12
13
14 #function signature of the imported functions
15 # finds all the objects with the given color and shape, returns a mask
        denoting the probability of object selection
16 filter(color, shape)
17
18 # finds the size of the structure struct_name that is formed with objects
        of the given type, returns the size of the structure (whose type is
        integer), arguments for this should be provided as kwargs
19 find_size(struct_name = str_name, objects = ObjSet)
20
21 # assigns the head to the location of the object
22 assign_head(at_obj_loc)
23
```

```
24 # moves the head in the given dir
25 move_head(dir)
26
27 # keeps the object obj at the head
28 keep_at_head(obj)
29
30
31 #Examples:
32 #Instruction: Move the green block to the left of the red dice
33 assign_head(at_obj_loc = filter(red, dice))
34 move_head(left)
35 keep_at_head(obj = filter(green, cube))
36
37 # Instruction: Find the size of the tower made of yellow legos
38 find_size(struct_name = tower, objects = filter(yellow, lego))
39
40 #Instruction: Find the size of the row made of orange cubes
41 find_size(struct_name = row, objects = filter(orange, cube))
42
43 # Instruction: Find the size of the column made of cyan cubes
44 find_size(struct_name = column, objects = filter(cyan, cube))
45
46 # Instruction: Move the green block to the left of the red dice and the
       yellow block to the top of the green block
47 assign_head(at_obj_loc = filter(red, dice))
48 move_head(left)
49 keep_at_head(obj = filter(green, cube))
50 assign_head(at_obj_loc = filter(green, cube))
51 move_head(top)
52 keep_at_head(obj = filter(yellow, cube))
53
54 # Instruction: Construct a row of green legos of length 3 to the right of
        the blue block
55 assign_head(at_obj_loc = filter(blue, block))
56 move_head(right)
57 row(length = 3, objects = filter(green, legos))
58
59
60 # Instruction: Construct a tower of size 3 using red cubes
61 tower(height = 3, objects = filter(red, cube))
62
63 # Instruction: Construct a row of size 5 using blue legos
64 row(length = 5, objects = filter(blue, lego))
65
66 # Instruction: Construct a column of size 6 using green die
67 column(length = 6, objects = filter(green, dice))
68
69 # Instruction: Place 3 green blocks so that one block is to the right of
        the other
70 green_blocks = filter(green, block)
71 keep_at_head(green_blocks)
72 move_head(right)
73 keep_at_head(green_blocks)
74 move_head(right)
75 keep_at_head(green_blocks)
76
77 # Instruction: Place 3 red legos on top of one another
78 red_legos = filter(red, lego)
79 keep_at_head(red_legos)
80 move_head(top)
81 keep_at_head(red_legos)
82 move_head(top)
83 keep_at_head(red_legos)
84
85
```

```
86 # CURRENT TASK
87 # Instruction: Construct tower of white cubes to the same height as
88 # existing tower of green die
89
90 # GPT-4s output
91 #First, we have to find the height of the tower of green dice,
92 #then construct a tower of white cubes of the same size
93
94 tower_size = find_size(struct_name = tower, objects = filter(green, die))
95 tower(height = tower_size , objects = filter(white, cube))
```

Listing 9: Prompting method for the task of constructing tower of white cubes to the same height as existing tower of green die

```
1  # Given a task you have to provide Python code for executing the task
2
3  # importing available functions
4  from spatial_directions import top, front, back, left, right
5  from primitives import assign_head, move_head, keep_at_head
6  # HEAD is a imaginary pointer keeping track of the current spatial
       location in consideration
7  from helpers import find_size, filter
8  from structures import row, column, tower
9  #function signature of the imported functions
10 # finds all the objects with the given color and shape, returns a mask
       denoting the probability of object selection
11 filter(color, shape)
12 # finds the size of the structure struct_name that is formed with objects
        of the given type, returns the size of the structure (whose type is
        integer), arguments for this should be provided as kwargs
13 find_size(struct_name = str_name, objects = ObjSet)
14 # assigns the head to the location of the object
15 assign_head(at_obj_loc)
16 # moves the head in the given dir
17 move_head(dir)
18 # keeps the object obj at the head
19 keep_at_head(obj)
20
21 #Examples:
22 #Instruction: Move the green block to the left of the red dice
23 assign_head(at_obj_loc = filter(red, dice))
24 move_head(left)
25 keep_at_head(obj = filter(green, cube))
26
27 # Instruction: Find the size of the tower made of yellow legos
28 find_size(struct_name = tower, objects = filter(yellow, lego))
29
30 #Instruction: Find the size of the row made of orange cubes
31 find_size(struct_name = row, objects = filter(orange, cube))
32
33 # Instruction: Find the size of the column made of cyan cubes
34 find_size(struct_name = column, objects = filter(cyan, cube))
35
36 # Instruction: Move the green block to the left of the red dice and the
       yellow block to the top of the green block
37 assign_head(at_obj_loc = filter(red, dice))
38 move_head(left)
39 keep_at_head(obj = filter(green, cube))
40 assign_head(at_obj_loc = filter(green, cube))
41 move_head(top)
42 keep_at_head(obj = filter(yellow, cube))
43
44 # Instruction: Construct a row of green legos of length 3 to the right of
        the blue block
45 assign_head(at_obj_loc = filter(blue, block))
```

```
46 move_head(right)
47 row(length = 3, objects = filter(green, legos))
48
49
50 # Instruction: Construct a tower of size 3 using red cubes
51 tower(height = 3, objects = filter(red, cube))
52
53 # Instruction: Construct a row of size 5 using blue legos
54 row(length = 5, objects = filter(blue, lego))
55
56 # Instruction: Construct a column of size 6 using green die
57 column(length = 6, objects = filter(green, dice))
58
59 # Instruction: Place 3 green blocks so that one block is to the right of
       the other
60 green_blocks = filter(green, block)
61 keep_at_head(green_blocks)
62 move_head(right)
63 keep_at_head(green_blocks)
64 move_head(right)
65 keep_at_head(green_blocks)
66
67 # Instruction: Place 3 red legos on top of one another
68 red_legos = filter(red, lego)
69 keep_at_head(red_legos)
70 move_head(top)
71 keep_at_head(red_legos)
72 move_head(top)
73 keep_at_head(red_legos)
74
75
76 # CURRENT TASK
77 # Instruction: Construct a tower of height 6 using red and blue blocks
       that are alternating
78
79 # GPT-4s output
80 # Python code:
81
82 # Define the red and blue blocks
83 red_blocks = filter('red', 'block')
84 blue_blocks = filter('blue', 'block')
85
86 # Start at the bottom and alternate building the tower
87 for i in range(6):
88     if i%2 == 0: # if the stack position is even
89         keep_at_head(red_blocks)
90     else: # if the stack position is odd
91         keep_at_head(blue_blocks)
92     if i != 5:  # if not at the top of the tower
93         move_head(top)
```

Listing 10: Prompting method for the task of constructing tower of alternating red and blue cubes

To find the size of a given structure in the given scene we define the function `find_size`, which takes the name of structure, all the objects in the initial scene, mask of the objects (a distribution over the objects based on the attributes), and the initial state. (we assume that this function has access to the semantics of all the concepts learnt so far, through a transition function). Algorithm 1 gives the pseudocode for the function find_structure. Below we provide a brief explanation for it.

1. First we assign our head to every block in the available blocks (line 6)

2. Then we begin constructing/visualizing the corresponding structure from that block beginning with a size of 1. (line 7)

3. For each structure created/visualized we compare the blocks moved for the structure creation with corresponding blocks originally present in the scene, and perform a matching between these blocks and a subset of the blocks originally present (line 11-26).

4. If we are able to find a mapping for each moved block, such that each mapped pair has an IoU greater than a threshold, we increase the next potential size to test by 1 (line 26-27).

5. The final size is the size corresponding to 2nd last iteration, before termination (line 28).

6. We return the maximum of all the possible structures that are found (line 34)

---

**Algorithm 1** Find Size

---

**Require:** $name$: structure name, $objs$: object list, $state$: initial state, $mask$: object mask
**Ensure:** Size of the maximum sized structure found
1: $found \leftarrow []$
2: **for** each $cand$ in $objs$ **do**
3:     $size \leftarrow 1$
4:     $curr \leftarrow state.copy()$
5:     **while** true **do**
6:         $new\_state \leftarrow assign\_head(cand)$
7:         $vis\_state, num\_mov, rew \leftarrow transition($
8:             $new\_state, name, [('size', size), ('objects', mask)])$
9:         $topk \leftarrow torch.topk(mask, num\_mov)$
10:         $possible \leftarrow true$
11:         $matches \leftarrow []$
12:         **for** each $idx$ in $topk$ **do**
13:             $possible \leftarrow (idx \ in \ objs)$
14:             $match\_ok \leftarrow false$
15:             **for** $m\_idx, obj$ in enumerate(state.state) **do**
16:                 $iou \leftarrow iou2d(vis\_state.state[idx],$
17:                     $state.state[m\_idx])$
18:                 $match\_ok \leftarrow (iou > 0.75)and$
19:                     $(m\_idx \ in \ objs)$
20:                 **if** $match\_ok$ **then**
21:                     $matches.append(m\_idx)$
22:                     break
23:                 **end if**
24:             **end for**
25:             **if** not $match\_ok$ **then**
26:                 break
27:             **end if**
28:         **end for**
29:         **if** $possible$ **then**
30:             $size \leftarrow size + 1$
31:         **else**
32:             $found.append((size - 1, matches))$
33:             break
34:         **end if**
35:     **end while**
36: **end for**
37: **return** max(size for size, match in $found$)

---

### D.4 DETAILS ON MCTS VARIANTS FOR PLAN SEARCH

Here we provide the details for 3 different plan search methods, that search over the space $\mathcal{A} = \mathcal{A}_c \cup \mathcal{A}_p$

- MCTS+*L*+*P*: This is the approach that we describe in section 4.2. For every concept say `Tower` $\in \mathcal{L}$ we have a corresponding set of macro action say `Make_Tower(3,`

objects) (*L*). Further we use a neural pruner $\pi_{\text{neural}}$ that outputs a primitive action $a_p^*$ (given current state and next expected state). We only consider the actions $\mathcal{A}_c \cup \{a_p^*\}$ during our search from the given state. This helps to reduce the effective branching factor and allows to search longer length plans within the same computational budget.

- MCTS+*L-P*: Here we do not use the reactive policy, therefore the branching factor for every node becomes $\mathcal{A} = \mathcal{A}_c \cup \mathcal{A}_p$.

- MCTS-*L+P*: We only search among the space of primitive actions i.e $\mathcal{A}_p$. Given a state $\tilde{s}_t$ and corresponding next expected state $s_{t+1}$ we greedily pick the action $a_p^* = \pi_{\text{neural}}(\tilde{s}_t, s_{t+1})$. This method is much more faster than the previous 2 methods as there is no explicit search. However the corresponding policy is trained only to output an action $a \in \mathcal{A}_p$ and lacks the ability to output modular plans composed of macro actions such as Make_Tower(3, objects) $\in \mathcal{A}_c$ (the action space $\mathcal{A}_c$ is increasing with time and the architecture of network needs to be changed accordingly). As a result the plans found are not modular and difficult to generalize. Further, the reactive policy is not trained to output reset_head(), store_head() as additional annotated data is required in order to train a classifier over them. This further decreases the space of grounded plans (and therefore corresponding generic programs) such a policy can represent. Training a reactive policy that can handle actions such as reset_head() and an action space $\mathcal{A}_c$ that grows with time is part of future work.

## D.5 GOAL-CONDITIONED PLANNING WITH LEARNT CONCEPTS

**Why is it difficult to hand encode a PDDL for our domain?** Most of the PDDL description of blocks world assume actions involving only the spatial relation onTop, which limits their applicability to describing different structures like *row* that need spatial relations like onRight. Further a single action might lead to varied effects/post-conditions based on the initial state. 16 gives 2 example of the same action moveOnTop(A, B) which would end up giving adding different number of spatial relations.

**Approach Overview.** Given an instruction $\Lambda$ = "Construct a staircase of magenta die having 3 steps", we first convert it into corresponding grounded task sketch $H_S^*$ = staircase(3, [3, 2, 1, $\cdots$]). Executing the corresponding program of staircase (by getting the semantics from the library $\mathcal{L}$) on the desired objects we get the expected final scene $S_f'$ in bounding box space. The initial scene $S_i$ and expected final scene $S_f'$ are converted into scene graph $SG_i'$ and $SG_f'$ (described in D.5). The relations between the task relevant objects in $SG_f'$ act as propositions/relations for goal check and the initial scene graph act as the initial state. Then a neuro-symbolic planner is used to obtain the optimal plan from the start state to a state that satisfies the goal. Below we also detail different aspects of the approach.

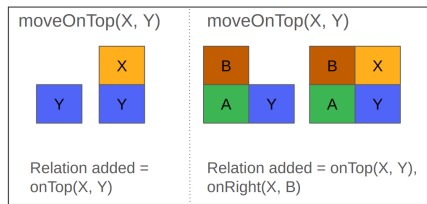

| moveOnTop(X, Y) | moveOnTop(X, Y) |
|---|---|
| Relation added = onTop(X, Y) | Relation added = onTop(X, Y), onRight(X, B) |

Figure 16: **Difficult to encode post-conditions**. Illustration of a domain where encoding a PDDL for direct planning is challenging.

**Scene-graph Extraction.** 0 gives the algorithm used for generating scene graph from a given scene (set of bounding boxes). Suppose we need to check whether there exists a relation of the form $(i, j, direction)$ i.e. block $i$ is in the direction $direction$ of block $j$, in a given scene. We first initialize the head at the position/bounding-box of block $j$ (line 7). Then we move the head in direction $direction$ (line 9). We claim that the relationship would exists if bounding box for block $i$ has IoU > 0.75 with the predicted_head (line 10-12).

---

**Algorithm 2** Get Scene Graph

---

1: **procedure** GET_SCENE_GRAPH($bboxes$) // bboxes describe the corresponding scene for which scene graph is to be found
2:     $spatial\_relations \leftarrow []$
3:     $directions \leftarrow \{$'right', 'front', 'top'$\}$
4:     **for** $i, bbox$ in $enumerate(bboxes)$ **do**
5:         **for** $j, other\_bbox$ in $enumerate(bboxes)$ **do**
6:             $final\_head \leftarrow bboxes[i]$
7:             $init\_head \leftarrow bboxes[j]$
8:             **for** $direction$ in $directions$ **do**
9:                 $pred\_head \leftarrow$ move_head$(direction, init\_head)$
10:                 $iou\_score \leftarrow IoU(final\_head, pred\_head)$
11:                 **if** $iou\_score > 0.75$ **then**
12:                     $spatial\_relations.append((i, j, direction))$
13:                 **end if**
14:             **end for**
15:         **end for**
16:     **end for**
17:     **return** $spatial\_relations$
18: **end procedure**

---

**Pre-conditions.** We define the following 2 preconditions (and learn their grounding) in order to ensure that the generated plans are physically possible.

1. `is-clear(blk, dir)`: We need to check whether a block blk has some free space in direction dir. For this we simply move the head in the direction dir with respect to the block blk. If the resulting position of head has 0 overlap with bounding boxes of all the other objects the predicate is True otherwise False.
2. `will-not-be-floating(pred_loc)`: We need to check whether the resultant/predicted location of an object on taking an action would be dynamically stable or not. The location would be stable if either it is on top of some already placed object or it is on the table surface. The former can be checked through the resultant scene graph itself (that is obtained by applying the algorithm 0), while for the later we train an *on-table* classifier. This would take as input a bounding box and predict whether the box is on the table or not. For training this classifier we use the dataset of pretraining phase. The blockwise positive and negative sample annotation can be done automatically by giving GPT-4 the corresponding scene graph and then querying which objects are on the table and which aren't. 17 gives an example. (Though we have not taken this approach for the complete dataset of 5k samples due to high cost).

**Actions.** We define the following two types actions

1. `place-random(blk)`: To place a block at a random free position on the table. For this we train a generative model (VAE) which learns the underlying distribution of bounding boxes for all the blocks that lie on the table. Given a scene we would sample position (bounding box) from this until we get a position that is not overlapping with the existing blocks in the scene. For training the VAE we assume that for every demonstration in the pretraining data, the first scene has all the objects randomly placed on the table (we could have also used the positive examples used for training on-table classifier).
2. `move(rel, blk1, blk2)`: This action corresponds to moving the blk1 in the direction rel of blk2, resulting in the addition of a relation (rel, blk1, blk2) in the set of spatial relations. This action is defined as a sequential composition of the actions `assign_head(blk2)`, `move_head(rel)`, `keep_at_head(blk1)` (blk1 is a one hot tensor for the corresponding block).

**Techniques and heuristic for efficient planning.** Since the action space for the planner could be $o(n^2)$, where n is the number of objects we adopt the following techniques to make planning scalable/efficient:

Figure 17: **Method to get annotation for training on-table classifier.**

1. Heuristic - We define the heuristic value h(s) for a state s, as the number of relations that are present in the goal but are absent in the scene graph corresponding to the state s. Even though this heuristic is not admissible (as it may over overestimate the cost to goal), it was found to work optimally in most of the cases.
2. Greedy-pruning - We assume that all the actions resulting in states with higher or same heuristic value would be of the form `place-random(blk)`. This means among the actions of the form `move(rel, blk1, blk2)` we only select those that lead to states with decreased heuristic value.
3. Relevant-object-set - Suppose O is the set of objects that are part of atleast one of the predicate in goal. We define O' as the transitive closure of O with respect to the relation $Related$ in the initial state $s_i$, where $SG(s_i)$ is scene graph for the initial state

$$Related(a, b, s_i) \iff \exists dir((dir, a, b) \in SG(s_i) \vee (dir, b, a) \in SG(s_i)) \qquad (9)$$

We assume O' is the relevant set of object for completing the task and actions that move any other object should not be taken.

## E  BROADER IMPACT

This work creates foundational knowledge in understanding human-like spatial abstractions. This work contributes towards the development of explainable and interpret-able learning architectures that may eventually contribute towards the development of embodied agents collaborating with and assisting humans in performing tasks. No negative impact of this work is envisioned.

## F  HYPERPARAMETERS, ARCHITECTURE DETAILS AND GROUND TRUTH CONCEPTS

### F.1  ARCHITECTURE FOR NEURAL MODULES

**Action Simulator:**

```
1  import torch.nn as nn
2
3  class ActionSimulatorNetwork(nn.Module):
4      def __init__(self, bbox_mode, hidden_size = 256):
5          super(ActionSimulatorNetwork, self).__init__()
6          self.bbox_mode = bbox_mode
```

```
 7          self.hidden_size = hidden_size
 8
 9          self.action_semantics_encoder = nn.Sequential(
10              nn.Linear(5, hidden_size),
11              nn.ReLU(),
12              nn.Linear(hidden_size, hidden_size),
13              nn.ReLU()
14          )
15          self.argument_encoder = nn.Sequential(
16              nn.Linear(5, hidden_size),
17              nn.ReLU(),
18              nn.Linear(hidden_size, hidden_size),
19              nn.ReLU()
20          )
21          self.decoder = nn.Sequential(
22              nn.Linear(hidden_size, hidden_size),
23              nn.ReLU(),
24              nn.Linear(hidden_size, 5),
25              nn.Tanh()
26          )
```

Listing 11: Action Simulator Network in PyTorch

**Reactive Policy($\pi_{neural}$):**

```
 1  import torch.nn as nn
 2
 3  class NeuralSearch(nn.Module):
 4      def __init__(self, action_space=6):
 5          super(NeuralSearch, self).__init__()
 6          self.action_space = action_space
 7          self.fc1 = nn.Linear(10, 256)
 8          # self.bn1 = nn.BatchNorm1d(256)
 9          self.bn1 = nn.Identity()
10          self.fc2 = nn.Linear(256, 256)
11          # self.bn2 = nn.BatchNorm1d(256)
12          self.bn2 = nn.Identity()
13          self.fc3 = nn.Linear(256, 256)
14          # self.bn3 = nn.BatchNorm1d(256)
15          self.bn3 = nn.Identity()
16          self.fc4 = nn.Linear(256, action_space)
```

Listing 12: Neural Search in PyTorch

**Random Position predictor** (for grounding of `place-random(blk)`):

```
 1  import torch.nn as nn
 2
 3  class VAE(nn.Module):
 4      def __init__(self, input_dim, latent_dim):
 5          super(VAE, self).__init__()
 6          self.input_dim = input_dim
 7          self.latent_dim = latent_dim
 8          # Encoder
 9          self.fc1 = nn.Linear(input_dim, 512)
10          self.bn1 = nn.BatchNorm1d(512)
11          self.fc2 = nn.Linear(512, 512)
12          self.bn2 = nn.BatchNorm1d(512)
13          self.fc3 = nn.Linear(512, 512)
14          self.bn3 = nn.BatchNorm1d(512)
15          self.fc4 = nn.Linear(512, 512)
16          self.bn4 = nn.BatchNorm1d(512)
17          self.fc51 = nn.Linear(512, latent_dim)  # Mean of the latent
      space
18          self.fc52 = nn.Linear(512, latent_dim)  # Log-variance of the
      latent space (log-var for numerical stability)
```

```
19
20          # Decoder
21          self.fc5 = nn.Linear(latent_dim, 512)
22          self.bn5 = nn.BatchNorm1d(512)
23          self.fc6 = nn.Linear(512, 512)
24          self.bn6 = nn.BatchNorm1d(512)
25          self.fc7 = nn.Linear(512, 512)
26          self.bn7 = nn.BatchNorm1d(512)
27          self.fc8 = nn.Linear(512, 512)
28          self.bn8 = nn.BatchNorm1d(512)
29          self.fc9 = nn.Linear(512, input_dim)
```

Listing 13: VAE in PyTorch

**On-table classifier** (for grounding of `will-not-be-floating(pred_loc)`:

```
1  import torch.nn as nn
2
3  class TableClassifier(nn.Module):
4      def __init__(self):
5          super(TableClassifier, self).__init__()
6          self.fc1 = nn.Linear(5, 16)
7          self.bn1 = nn.BatchNorm1d(16)
8          self.fc2 = nn.Linear(16, 16)
9          self.bn2 = nn.BatchNorm1d(16)
10         self.fc3 = nn.Linear(16, 16)
11         self.bn3 = nn.BatchNorm1d(16)
12         self.fc4 = nn.Linear(16, 1)
13         self.bn4 = nn.BatchNorm1d(1)
14         self.sigmoid = nn.Sigmoid()
```

Listing 14: Table Classifier in PyTorch

### F.2 HYPERPARAMETERS USED IN EXPERIMENT

As indicated in A.3 for the purpose of generalization through multiple candidate plans (from 1 demonstration) we chose the top-k plans (as measured by overall IoU achieved). The k chosen for all our experiments involving MCTS was 5. (The performance of our best approach was found to be the same for k=5 to 20). For every plan we obtain 3 programs from GPT-4 by re-prompting it 3 times with the same input prompt (with temperature > 0). From the pool of these 3*k programs we chose the one with highest IoU reward by running each of them on the given demonstration. The discount factor kept for our search is $\gamma = 0.95$, and unless explicitly specified the number of expansions steps used = 5000.

### F.3 GROUND-TRUTH INDUCTIVE CONCEPTS

```
1  ######
2  # row
3  def row(length, objects):
4      for i in range(length):
5          keep_at_head(obj = objects)
6          move_head(dir = "right")
7
8  ######
9  # tower
10 def tower(height, objects):
11     for i in range(height):
12         keep_at_head(obj = objects)
13         move_head(dir = 'top')
14
15 ######
16 # column
17 def column(size, objects):
```

```
18      for _ in range(size):
19          keep_at_head(obj = objects)
20          move_head(dir = 'front')
21
22  ######
23  # staircase
24  def staircase(steps, objects):
25      for step in range(1, steps+1):
26          tower(height = step, objects = objects)
27          move_head(dir = 'right')
28
29  ######
30  # inverted_row
31  def inverted_row(num, objects):
32      for i in range(num):
33          keep_at_head(obj=objects)
34          move_head(dir='left')
35
36  ######
37  # inverted_column
38  def inverted_column(size, objects):
39      for _ in range(size):
40          keep_at_head(obj = objects)
41          move_head(dir = 'back')
42      return None
43
44  ######
45  # inverted_staircase
46  def inverted_staircase(steps, objects):
47      for step in range(1, steps+1):
48          tower(height = step, objects = objects)
49          move_head(dir = "left")
50
51  ######
52  # diagonal_135
53  def diagonal_135(length, objects):
54      for i in range(length):
55          keep_at_head(obj = objects)
56          move_head(dir = 'front')
57          move_head(dir = 'left')
58      return
59
60  ######
61  # diagonal_315
62  def diagonal_315(length, objects):
63      for i in range(length):
64          keep_at_head(obj = objects)
65          move_head(dir = 'back')
66          move_head(dir = 'right')
67      return
68
69  ######
70  # diagonal_225
71  def diagonal_225(length, objects):
72      for _ in range(length):
73          keep_at_head(obj = objects)
74          move_head(dir = 'back')
75          move_head(dir = 'left')
76
77  ######
78  # diagonal_45
79  def diagonal_45(length, objects):
80      for _ in range(length):
81          keep_at_head(obj = objects)
82          move_head(dir = 'front')
```

```python
83          move_head(dir = 'right')
84
85  ######
86  # boundary
87  def boundary(size, objects):
88      row(length=size-1, objects=objects)
89      for _ in range(size-1):
90          move_head(dir = 'right')
91      move_head(dir = 'front')
92
93      column(length=size-1, objects=objects)
94      for _ in range(size-1):
95          move_head(dir = 'front')
96      move_head(dir = 'left')
97
98      inverted_row(length=size-1, objects=objects)
99      for _ in range(size-1):
100         move_head(dir = 'left')
101     move_head(dir = 'back')
102
103     inverted_column(length=size-1, objects=objects)
104     for _ in range(size-1):
105         move_head(dir = 'back')
106     move_head(dir = 'right')
107
108 ######
109 # arch_bridge
110 def arch_bridge(height, objects):
111     staircase(steps = height, objects = objects)
112     move_head(dir = 'left')
113     inverted_staircase(steps = height, objects = objects)
114     return
115
116 ######
117 # x-shaped structure
118 def x(size, objects):
119     diagonal_45(length = size, objects = objects)
120     move_head(dir = 'back')
121     diagonal_315(length = size, objects = objects)
122     move_head(dir = 'left')
123     diagonal_225(length = size, objects = objects)
124     move_head(dir = 'front')
125     diagonal_135(length = size, objects = objects)
126
127 ######
128 # pyramid
129 def pyramid(height, objects):
130     for i in range(height):
131         row_length = (height * 2) - (i * 2) - 1
132         row(length = row_length, objects = objects)
133         if i != height - 1:
134             move_head(dir = 'top')
135             move_head(dir = 'right')
136 ######
```

Listing 15: Definition of inductive concepts

## G  COMPUTATIONAL REQUIREMENTS: DETAILS

All our experiments were run on a server with the following machine specifications.

**CPU Specification:**

| Specification | Value |
|---|---|
| Architecture | x86_64 |
| CPU op-mode(s) | 32-bit, 64-bit |
| Address sizes | 46 bits physical, 57 bits virtual |
| Byte Order | Little Endian |
| CPU(s) | 112 |
| On-line CPU(s) list | 0-111 |
| Vendor ID | GenuineIntel |
| Model name | Intel(R) Xeon(R) Gold 6330 CPU @ 2.00GHz |
| CPU family | 6 |
| Model | 106 |
| Thread(s) per core | 2 |
| Core(s) per socket | 28 |
| Socket(s) | 2 |
| Stepping | 6 |
| CPU max MHz | 3100.0000 |
| CPU min MHz | 800.0000 |
| BogoMIPS | 4000.00 |

**GPU Specification:**

| Specification | Value |
|---|---|
| **GPU 1** | |
| Description | VGA compatible controller |
| Product | Integrated Matrox G200eW3 Graphics Controller |
| Vendor | Matrox Electronics Systems Ltd. |
| Physical ID | 0 |
| Bus Info | pci@0000:03:00.0 |
| Logical Name | /dev/fb0 |
| Version | 04 |
| Width | 32 bits |
| Clock | 66MHz |
| Capabilities | pm vga_controller bus_master cap_list rom fb |
| Configuration | depth=32 driver=mgag200 mingnt=16 |
| Resources | irq:16 memory:91000000-91ffffff memory:92808000-9280bfff memory:92000000-927fffff memory:c0000-dffff |
| **GPU 2** | |
| Description | 3D controller |
| Product | GA102GL [A40] |
| Vendor | NVIDIA Corporation |
| Physical ID | 0 |
| Bus Info | pci@0000:17:00.0 |
| Version | a1 |
| Width | 64 bits |
| Clock | 33MHz |
| Capabilities | pm bus_master cap_list |
| Configuration | driver=nvidia latency=0 |
| Resources | iomemory:21000-20fff iomemory:21200-211ff irq:18 memory:9c000000-9cffffff memory:210000000000-210ffffffff memory:212000000000-212001ffffff memory:9d000000-9d7fffff memory:211000000000-211ffffffff memory:212002000000-212041ffffff |
| **GPU 3** | |
| Description | 3D controller |
| Product | GA102GL [A40] |
| Vendor | NVIDIA Corporation |

| Physical ID | 0 |
|---|---|
| Bus Info | pci@0000:ca:00.0 |
| Version | a1 |
| Width | 64 bits |
| Clock | 33MHz |
| Capabilities | pm bus_master cap_list |
| Configuration | driver=nvidia latency=0 |
| Resources | iomemory:28000-27fff iomemory:28200-281ff irq:18 memory:e7000000-e7ffffff memory:280000000000-280ffffffff memory:282000000000-282001ffffff memory:e8000000-e87fffff memory:281000000000-281ffffffff memory:282002000000-282041ffffff |

**Time Required:** The time required for pretraining phase of all the neural modules is around 36 hours. For learning of inductive concepts the time taken varies from 5 minutes to 1 day depending on the search method used and the specific set of hyperparameters. However for our best approach we get the maximum performance in approx 12 minutes. Time taken for our approach during inference is less than 2 minutes per dataset.

