# OpenReview forum: "Sketch-Plan-Generalize: Learning Inductive Representations for Grounded Spatial Concepts"
_ICLR.cc/2025/Conference — ICLR 2025 Conference Withdrawn Submission_

### Official Review · Reviewer_D7es · 2024-11-01

**Soundness:** 2
**Presentation:** 3
**Contribution:** 2
**Rating:** 5
**Confidence:** 4

**Summary:**

This paper introduces an LLM-based three-step inductive program synthesis pipeline:
1. Convert the natural language into a function signature with LLM (sketch),
2. Do visual grounding, and do MCTS using the available concepts and grounded actions—check if each expanded state matches with the goal image (plan),
3. If the goal is found, give the action sequence to the LLM to create a new program (generalize).

The pipeline uses an inductive formulation of the function signature to allow for structures of any size and property (e.g., color). Experiments done in simulation with simple architectures show better results when compared with alternative LLM prompts.

**Strengths:**

- Overall, I like the presentation of the paper. Sections 1, 2, and 3 clearly outline the problem, give the gist of the method, and formulate the problem.
- I like the idea of learning new programs/concepts by essentially decomposing examples into their function signatures plus specific instantiations (i.e., size, color, position). The overall pipeline—to the best of my knowledge—is novel and can potentially be improved.
- While the given examples are a bit limited in terms of using previous concepts or inductive logic—which is the major point that draws me to the borderline—the method seems to be general and potentially can be further improved. I believe this paper might be of interest to the learning community, in particular those interested in program synthesis.

**Weaknesses:**

- Even though the method seems quite generic, the examples of generalization and new concept learning based on previous concepts are quite limited. For instance, in Lake 2015, there were many examples showing new characters using primitive structures, etc. My expectation was high while I was reading the first sections as I thought the formulation is quite generic and should allow for structures with 50 blocks or very new concepts with different types of blocks. Either the selected experiment set allowed only for these examples, or there are some fundamental limitations—which are not mentioned—that prevent the pipeline from scaling to such numbers.
- There is always the question of whether the pipeline will scale to many different concepts when an LLM plays a part—no easy way to verify or falsify it.

**Questions:**

- Can we construct something like:
```
:..:
:..:
:..:
:..:
:..:
:..:
```
where each inductive step consists of a composite object (here, there are five composite objects, each having two cubes on top of a long block)?
- How about structures where there are possibly more than one controlling variable, e.g., a rectangle with height and width?
- This is rather a personal opinion/question. In Eq. 1, position feels like a parameter regarding the task, not necessarily specific to the formulation. E.g., we could have included many different parameterizations that are known to the agent. So, maybe, one can give it a generic name and collect all such parameters inside that. Likewise, ‘size’ can possibly refer to things that are not physical sizes.
- Why C_{k’} in the composition? Does it have to be strictly a substructure of C_k? Also, can the composition part be optional? Maybe a running example would be better.
- The first term (the loss part) in Equation 2 is not clear. Does it measure the joint probability of that hypothesis and the frames, or the likelihood of those frames given the hypothesis?
- Line 917, “assuming there is only one unknown concept”, I think this is an important assumption and should be included in the main text.
- Line 919, “If the unknown concept is a primitive concept, a concept embedding is learned through backpropagation”, does this mean we fine-tune the LLM? Also, it’s not very clear what those parameters (\theta_s, \theta_p, \theta_g) are part of. LLM’s parameters?
- I know the space is tight but it would be better if you can detail the grounding procedure in Sec. 5.1, at least in the appendix.
- I believe MCTS assumes that there is a forward model allowing intermediate states to be computed, which should be listed in Sec. 3.
- Line 311, “Additionally, to prune primitive actions, we train a reactive policy \pi_{neural} which, given the current state \tilde{s} t and the next expected state s {t+1} (from the demonstration), outputs one of the primitive actions a* t \in \mathcal{A}_p”. I don’t quite understand the reasoning here. Having a reactive policy would give us a single action output; I’m not sure if this should be called pruning. Wouldn’t removing all those primitive actions aggressively limit the search space of MCTS such that we may fail to find a solution? I see that the number of actions increases when we include macro actions, but it shouldn’t be too much at least for the examples given in the paper?
- Line 876 typo, Scalability
- line 887, did you mean Listing 6 instead of 15?
- Line 324, “Appendix C.2, Fig. 2, A.4 details the curriculum used for concept learning”, should this be A.2? Please recheck all the links because I believe they might be mixed in general.
- As there are lots of parameters regarding the experiments, models, baselines, etc., my key question to answer in Sec. 7 would be what kind of inductions that this pipeline can model? I think this is one of the key points of the paper, and it would have been more informative to report such examples. Also, maybe, showing how the system can build up concepts of increasing complexity. Towers -> Staircase + X mark -> A real pyramid? Because I’m not sure whether the comparisons make too much sense—not that I don’t understand, but I can’t easily translate those results. For instance, one can expect at least the word ‘inductive’ to appear in those prompts (Listing 8 and Figure 13) if you want to test if LLMs and VLMs can do inductive reasoning. I **do** appreciate the effort—but I’m not sure if it tells too much.

---

### Official Review · Reviewer_LfTc · 2024-11-02

**Soundness:** 1
**Presentation:** 2
**Contribution:** 2
**Rating:** 3
**Confidence:** 3

**Summary:**

Problem statement. The paper proposes a system to produce representations that are useful in inductive reasoning for agent learning. They define inductive concepts as those that can be described recursively or as a composition of smaller structures.

Method. Given a task command, their method begins by predicting a programatic "sketch" of the policy that is not grounded (in that generated functions may not be defined).  For planning, a set of primitive actions and composed actions define the action space and the search is guided by the reward mIoU between attained and demonstration representation. In the final generalization step, these grounded plans are used to create an executable python program.

Evaluation. On a custom(?) simulation environment, the method is evaluated on three datasets that test generalization of simple structures. Compared to end-to-end learning approaches and zero-shot code generation models, their method produces the highest mIoU between generated and target structures.

**Strengths:**

The novelty of the approach is a strength of the paper. I appreciate the introduction of inductive representations into the standard code generation pipeline. The paper also provides a well described dataset that ablates pre-training knowledge of structure labels (e.g., reversing tower to rewot). On this dataset (Dataset II) the advantages of their approach compared to zero-shot code generation approaches is more evident (see Table 3). As structures get more complex and out-of-distribution from the training set of the underlying LLM, the SPG approach would probably be even stronger.

**Weaknesses:**

The biggest weakness of this paper is that it's hard to replicate the results and situate them in the context of other work. I couldn't find enough information about the simulator used to create the dataset. For example, in the Corpus section the paper says "A corpus is created using a simulated Robot Manipulator assembling spatial struc- tures on a table-top viewed by a visual-depth sensor." There is neither a citation for the simulator used nor a precise description of the method used to generate the datasets. This makes it very difficult to place this contribution in the context of other work, such as code-as-policies. It would be useful for the authors to (1) provide additional details about the evaluation protocol in the paper and either (2a) also compare on environments used by past work, such as code-as-policies, or (2b) sufficiently justify why a new evaluation protocol is needed. I am also worried that the evaluation protocol is quite narrow. This could be improved by broadening the set of structures evaluated substantially or by stress testing the approach in a real-world robotics task.

Notes:
- The notation for inductive representations is different in the text and in figure 2. One uses **H** and another uses **\mathcal{H}**

**Questions:**

How sensitive is the mIoU metric to the representations provided within the demonstration set? It's not obvious to me why this wouldn't limit the generalizability of learned plans.

---

### Official Review · Reviewer_ZzaC · 2024-11-03

**Soundness:** 1
**Presentation:** 1
**Contribution:** 2
**Rating:** 3
**Confidence:** 4

**Summary:**

This work is devoted to the development of a neurosymbolic method for generating representations of complex concepts that may consist of several elements. Such a representation is generated at the Sketch stage of the proposed SPG framework based on the so-called visual-grounding module relying on a visual extractor, concept-embedding module, and “quasi-symbolic executor”. At the output of this stage, some functional description is generated which includes predefined functions. Then, the Plan stage runs the MCTS planner to generate a sequence of actions that should correctly predict the sequence of frames available in the set of demonstrations of the trained concept construction. Finally, the Generalize step uses GPT-4 to generate Python code that describes a sequence of primitive steps of constructing a given concept in a general way. The authors perform an experimental comparison of the proposed approach with the conventional neural network planner StructDiffusion and with GPT-4(V) as a pre-trained baseline. The accuracy of concept description generation, generalizability to structure sizes, and robustness to concept name changes are shown. Problems for constructing multiple structures from blocks are used throughout.

**Strengths:**

The problem of concept generation and functional description by demonstrations, which the authors address, has more relevance to the construction of general-purpose planners. To the best of my knowledge, using MCTS to generate explanatory actions for the purpose of passing as examples to a language model has not been proposed before.

**Weaknesses:**

The paper as a whole is written in a complicated language, there are too many references to the appendix for important details of the method. The text is oversaturated with notations that are not always used later and not all are fully defined (for example, there is no definition of /eta_theta in formula 1). At several points, the authors refer to several papers (in an “akin to” style) when they talk about reusing some modules and it is not clear exactly which approach they are reusing. Overall, it is very difficult to understand the exact relationship of all the modules to each other and the superficial diagram in Fig. 2 does not make it very easy to understand the details. Specific shortcomings:
1. The most unclear place is the work of the Sketch module, where the very important step of grounding concepts takes place. The authors' direct phrase “The task sketch thus obtained is then grounded on the input scene using a quasi-symbolic visual grounding module akin to” is as unspecific as possible. Do the authors use some ready-made approach? Which of the three described? What is the basis of the pre-trained model? LLM or VLM? What is a “quasi-symbolic executor”? Is the resulting task sketch somehow checked (executed) for validity?
2. At the same step a visual extractor (ResNet-34 based) is used, which means that it is pre-trained on some dataset to form bounding boxes and that the set of concepts is known in advance. This, of course, limits the whole approach very much.
3. In the same step, the authors specify that “disentangled representations for visual concepts” are formed. However, disentangledness is a special property of latent representations that needs to be demonstrated, including showing that the extracted features are interpretable. It is not clear how vector representations of such features are associated with textual names.
4. The authors propose to use MCTS to generate an action plan that gives the ability to obtain the same states from some initial states that are in the demonstrations. It is not clear, do the primitives used give discrete actions? If continuous (which makes sense), how does MCTS handle continuous actions? If discrete, what is the discretization and how large does the search space get in general?
5. The use of “reactive policy” greatly simplifies the approach and generally questions the need for such overcomplication with MCTS, perhaps a simple tree search without reward would suffice. How long do the constructed plans turn out to be?
6. The Generalize phase is described as superficially as possible and even in the appendix it is hard to see how complex the LLM's work is if it is given a prompt as input (including “Rather than getting a single plan from the plan search we get the top k plans”), which largely already contains the correct answer and the necessary complex structure of the type concept and call cycle.
7. It is not at all clear what is even trainable in this approach. The authors indicated that there are learnable parameters \theta_S, \theta_P, \theta_G, but which ones are actually tuned in the training phase? I guess by \theta_P you mean the counters in MCTS or value function parameters? Either those are parameters of the reactive policy, but it is pre-trained. \theta_G must be the parameters of GPT-4, but it is also pre-trained. It is not at all clear with the parameters of \theta_S, because it does not specify at all what model is used.
8. The paper introduces formalization in the form of equations (1) and (2), but this only complicates the reading and raises more questions. What is the Loss Loss function used in formula (2)? What is the Exec function? There is no notation for \eta_theta. In fact, the authors use ready-made modules (like MCTS or pre-trained LLMs like GPT-4), for which there is their own formalization of their work, and it does not fit into the one introduced by the authors.
9. The experiments were conducted for a very narrow task with a very small set of tasks, a small vocabulary, and number of concepts. It is very likely that with proper tuning of the prompts and examples, models like GPT-4 will handle them better than the basic simple prompts used by the authors.

**Questions:**

A few questions from the Weaknesses section:
1) Do the authors use any off-the-shelf approach in the Sketch phase? Which one of the three described? What is the underlying pre-trained model there? LLM or VLM?
2) In MCTS, if the actions are discrete, what is the discretization and how large is the search space in general?
3) What is the Loss Loss function used in formula (2)? What is the Exec function?
4) It is not at all clear what is trained in this approach exactly at runtime without considering the pre-training of some components.

---

### Official Review · Reviewer_wWgV · 2024-11-03

**Soundness:** 3
**Presentation:** 2
**Contribution:** 3
**Rating:** 5
**Confidence:** 4

**Summary:**

This paper aims to learn generalized inductive programs for spatial concepts (like building staircases and pyramids), given a small number of demonstrations. The proposed method is a sequence of steps named Sketch-Plan-Generalize (SPG). First, a Large Language Model (GPT-4) is used to parse natural language instructions like “construct a staircase of size 4 using yellow cubes” and generate a program signature. Then, a mix of neural and symbolic methods are used to convert the program signature into a function call which executes the instruction in the environment. If such a function is not available in the library (when demonstrating a new concept), Monte Carlo Tree Search (MCTS) is used to plan a sequence of actions (program/function calls), from the available mix of predefined and learnt programs. Further, an LLM is prompted to generate a new program that outputs the planned sequence of actions (programs). This newly generated program is added to the library. The entire process is hypothesized to generalize a given demonstrated concept and build more complex programs as a composition of existing ones. Experiments are performed on a simulated environment with colored lego blocks and dices. A tractable and carefully designed hierarchy of building tasks (row, tower, staircase, …, etc) are chosen from a base set of programs (move_head, keep_at_head, … etc) and used for evaluation. The proposed method is shown to learn concepts and generalize much better than selected baselines. Appropriate ablations are conducted to show the efficacy of their algorithmic choices.

**Strengths:**

1. Originality
- Inductive Concepts: LLMs and other data-driven AI models consistently fail to generalize to inductive concepts. As humans it is puzzling how we are so good at it and yet machines fail to see the inductive pattens and extrapolate. This paper takes a programmatic approach to mitigate this problem, which I believe, is a fresh direction for the AI community, given the criticism around LLMs.
- Inductive Problem Representation: The formal representation of a spatial concept generator in equation $(1)$ page 4, containing induction, composition and base term is a helpful compression of the problem. I find the SPG pipeline built on top of this recursive expression conceptually neat and useful to the readers. I require some minor clarifications, however (see questions section).

2. Quality
- Good Design Choices: Effective use of LLMs. Using GPT-4 to absorb the complexity of natural language task instruction and code generation is well justified. They are shown to perform well in these tasks [1], [2].
- Relevant Ablations: The use of MCTS instead of an LLM planner at the “Plan” stage was justified with an Ablation in Section 7 and Figure 5 in Page 9. Different variants of MCTS were ablated to show the need for concept library and pruning. The authors acknowledged the branching factor issue in MCTS with a primitive action policy which is justified from the ablation.

3. Clarity
- Presentation: Even though the entire approach is quite complex with several moving parts, the authors do a thorough presentation, making it easier to comprehend and visualize the approach and results. Figure 2 and Supplementary Figure 12 are especially helpful.
- Detailed appendix: The authors provide a very informative and reproducible Appendix with details regarding code, LLM prompting, SPG sub-components and MCTS planning.

4. Significance
- Motivation and conceptual problem: Learning inductive concepts like building recursive structures and mathematical constructs (multiplication is repeated addition) is an important problem that LLMs and other purely data-driven algorithms do not generalize. I believe that the programmatic approach taken in SPG is a good direction towards combining neural and symbolic methods. Both the problem and the general direction of solution would be of high interest to the relevant community.
- Few-shot learning from demonstrations: Recent hype around ARC-AGI style problems have highlighted the need for in-context learning and rapid generalization. The paper attempts to solve rapid learning and generalization of inductive concepts that can potentially help develop generalist agents.

**Weaknesses:**

1. **Novelty**: One of my biggest concern in the paper is that the authors have not made it clear what exactly their scientific contribution is in the paper, and have not placed it appropriately in the prior work section, even though relevant prior works are cited. Inductive representations have been tackled recently in [3] on a more relevant ARC-AGI problem. Several popular works augment libraries to generate code with LLMs, and some even ground the programs on to real robots [2] [4]. Planning with abstract programs/concepts is also explored in [5]. I see the value in their approach but it is unclear in their presentation where this value stands when compared agains prior work. Not all components in SPG are novel in themselves. I believe it is the responsibility of the authors to highlight and claim their novel contribution, and help the community assess their knowledge addition, in light of existing work.
2. **Physically Grounded concept learning**: Adding on to (1), the authors make the claim that their approach is different from prior work on concept learning and robot instruction following (page 2, 3) because they physically ground their inductive programs on robots. This is not a fair claim in my opinion because they use an oracle simulator and assume the availability of a primitive set of actions such as "place object at a location" and "move robot head to a location" that need not be learnt. It is not clear which of the primitives are learnt from scratch with data and which of them are assumed to be available off-the-shelf. Also, Inductive learning of concepts and grounding them in robotic applications seem like two very different problems. I am not sure why both these problems were tackled at the same time - this can be clarified.
3. **Choice of Environment/Simulator**: I work in reinforcement learning and neuro-symbolic algorithms, but have not come across the environment authors used. It is also unclear why the given environment was used instead of existing ones with benchmarks that would have made it easier to compare and reproduce [7] [8] [9]. Please correct me here if I am mistaken, and if the environment is already a standard benchmark. If not, did the authors custom build the environment for the purpose of this paper? Is it chosen because no other benchmarked environment made it possible to demonstrate inductive concept learning? And is the chosen environment/simulator open sourced?
4. **Problem Scope**: Adding on to (1) and (2), the authors claim on page 2 that - “In contrast, this paper focus on learning a representation of specific class of higher-order spatially-grounded concepts, namely those possessing a notion of induction resulting in the construction of a structure” - this reduces the scope of SPG’s application to a very small set of problems. Which other relevant problem/domain/environment would the authors recommend to use, if I wanted to try and run SPG? This could be a small discussion in the conclusion/future work sections.
5. **Evaluation and Baselines**: I find the baselines to be fair and relevant for the task. However, since the environment and the tasks are carefully designed, it is difficult to claim that SPG algorithm is effective wherever inductive/compositional reasoning is required. An additional baseline to help verify this could be “code as policies” [2]. How would it fare against SPG if [2] is exposed to visual and robot grounding primitives + code for all demonstrations, without the inductive reasoning capability. What would happen in environments like Furniture Bench manipulation [9] which uses compositional components like legs of chair/table/desk, but also requires a more robust robot learning algorithm to perform primitive actions like tightening the screw (also inductive to some degree) or flipping the table with noisy control? More details on my exact evaluation request below.
6. **Learning of Visual concepts**: It is unclear to me, from both the discussion on page 5 and page 9 (last paragraph) how the “visual grounding” process works. Please explain. Would it be possible to extend the “Visual Grounder” block in the flow diagram as seen in Figure 7 (Supplementary)? Also, while the Visual Grounder is mentioned in the “Sketch” part of the pipeline (Page 5 last paragraph), it is represented in the “Plan” block in Figure 2. Please correct this.
7. **Other learnable modules**: There are several components in the entire SPG pipeline and it is unclear which of them are assumed to be available a priori, and which of them are learnt with data specific to the environment. Can we have a figure that clearly mentions all the learnable and oracle components in SPG and how they connect within the entire framework?

**Questions:**

### Request for Clarification and Additional Baselines

I explicitly highlight the clarifications and details I need to make a final call regarding the scores. The below concerns extend the points mentioned in weakness section respectively, and are the reasons why I lean towards a weak reject at this point. Authors, please directly address the issues below. **I am willing to increase my score towards an accept if these concerns are addressed appropriately during the discussion phase**. I have gone through the supplementary materials for relevant figures and explanations, but please point out the exact page/table/figure, if I have missed a detail that’s already provided in the paper or supplementary section.

1. Novelty: Not all components of SPG is novel and several such pipelines exist, that distill concepts and generate code with LLMs. Please make the contribution explicit in sections 1 and 2. Additionally, please re-assess the baselines and results in light of the novel contribution.

2. Please clarify if and why SPG can claim to ground inductive concepts in physical space, despite the use of simulator for planning and off-the-shelf primitive actions. If there are learnable components that make physical grounding more than executing a few pre-defined function calls, please mention that clearly.

3. Choice of Environment: Discuss either in the Evaluation Setup section or the Appendix regarding why the particular environment was chosen. Additionally, clarify the following either in the paper or in response to this review - Is the environment benchmarked and/or open-sourced? If not, did the authors custom build the environment? Can this be compared against other existing environments/simulator?

4. If I were to use and run SPG on one other open-sourced environment/problem, which environment/problem would the authors recommend? I am not asking for actual experiments here, but rather a discussion to learn where else would researchers benefit from using SPG over the baselines (purely neural/direct code output from LLMs) and other approaches.

5. Evaluation and Baselines: It is difficult to claim that SPG is an effective approach for grounded inductive/compositional pick-place/manipulation tasks. To test this out I request the following clarifications and experiments:
    - SPG uses a curriculum to learn from simple to complex problem. Are all the baselines exposed to the same curriculum of tasks for a fairer comparison?
    - Essentially, I would like to evaluate if explicit inductive programming+MCTS planning (the “Plan” step) is necessary to solve the structure building problems, or weather this combinatorial complexity could also be absorbed by GPT-4. To evaluate this, the authors can have one additional baseline - A GPT-4 code generator, which takes as inputs (1) task described in natural language, (2) visual grounding information/functions, (3) action primitives/functions (4) **The previously generated code from demonstrations in a curriculum** and directly outputs a final program that uses existing primitives and prior code to solve the newly described task. The approach is essentially same as prior work [2] with some additional information from code and demonstrations and this baseline does not have an explicit MCTS to help GPT inductively compose. I believe this experiment is fair and easy to evaluate - it would also show (if) why induction + MCTS might be necessary.

    - I want to reason about SPG's effectiveness on more realistic tasks and benchmarks **without additional experiments** on new environments, since that would not be feasible during the rebuttal. Furniture bench [9] has a simple problem description: Given some parts, assemble them into valid furnitures. It is a solid robotics benchmark for long horizon manipulation and is currently challenging to solve. Can the authors provide a hierarchy of programs/structures that would be required (similar to Appendix Figure 12 in the paper) to assemble all of the furnitures (~8) in Furniture Bench? How many simple and complex structures would be required? and would the MCTS planning in SPG perform well in the situation? How would SPG keep track of semi-assembled parts that are not necessarily inductive, but still a composition of existing primitive parts. You may assume a basic set of primitive actions "go-to-object" and "pick-object" for the robot. Please note that I am not asking the authors to solve Furniture Bench with SPG, but rather reason if SPG can be used in such scenarios.

6. Learning of Visual concepts: Exactly as mentioned in Weakness section.

7. Other Modules: Can we have a figure that clearly mentions all the learnable and oracle components in SPG and how they connect within the entire framework?

### Other concerns and comments (Not counted towards my scoring)

1. Language and Writing: There is room for improvement in the language and grammar used throughout the paper. Some parts were really difficult to parse. Examples I found:
    - Page 2 related works: “In contrast, this paper focus on learning … ” → **focuses**
    - Same paragraph: “such a search to the assessing the physical construction” → **to assess the physical construction**
    - Page 3, paragraph 1: “… works for synthesize efficient.. ” → **to synthesize**
    - Page 5, Section 5: “… modeling an inductive spatial concept modeling structures whose …” → **???**
    - Same paragraph: “Alternatively, neural methods attempting to predict action sequence to attain the assembly are challenged by continual setting where concepts can increase over time building on previously learnt ones but are resilient to noise.” → Sentence is too convoluted, please break it down.
2. Equation $(1)$ clarification: What is the symbol $\circ$ between Induction, Composition and Base terms? Is it multiplication? Composition?
3. Please add an index at the beginning of the Appendix. It is hard to figure out the structure of content in the appendix and look for relevant information.
4. As an aside, my opinion is that the neuro-symbolic research community would have benefited far more if the authors chose a subset of ARC-AGI [7] that relied on inductive reasoning without the need for grounding, OR a more realistic robotics benchmark like [9] with grounding. I would like to encourage the authors to try and apply SGP to these challenging tasks in the future.
5. No explicit reproducibility statement: Although, the authors provide copious code snippets and experiment details, I’d encourage the authors to commit to reproducibility of their work with code/environment release.


### Review Summary
Overall, I would not consider sketch-plan-generalize a groundbreaking idea, since it contains a mix of several existing and well established approaches, and is applied on some carefully designed experiments. That said, I think there is a lot of value in their approach. The paper attempts to solve inherently inductive problems in AI without having to re-learn for each instance. This kind of solution with strong generalization capacity might be of interest to concept learning, neuro-symbolic AI, planning and robotics community. I believe that an AGI system in the near future, must have test time planning and inductive generalization built into it. The authors work with a small problem space, but do a thorough job of accounting their code and experiments in the supplementary section to help reproduce their results.

### References
[1] Josh Achiam, Steven Adler, Sandhini Agarwal, Lama Ahmad, Ilge Akkaya, Florencia Leoni Aleman, Diogo Almeida, Janko Altenschmidt, Sam Altman, Shyamal Anadkat, et al. Gpt-4 technical report. *arXiv preprint arXiv:2303.08774*, 2023

[2] Jacky Liang, Wenlong Huang, Fei Xia, Peng Xu, Karol Hausman, Brian Ichter, Pete Florence, and Andy Zeng. Code as policies: Language model programs for embodied control, 2023.

[3] Wang, Ruocheng, et al. "Hypothesis search: Inductive reasoning with language models." *arXiv preprint arXiv:2309.05660* (2023).

[4] Grand, Gabriel, et al. "Lilo: Learning interpretable libraries by compressing and documenting code." *arXiv preprint arXiv:2310.19791* (2023)

[5] Liu, Weiyu, et al. "Learning Planning Abstractions from Language." *The Twelfth International Conference on Learning Representations*

[7] https://github.com/fchollet/ARC-AGI

[8] https://github.com/mila-iqia/babyai

[9] https://clvrai.github.io/furniture-bench/

---

### Note · Authors · 2024-12-12

I have read and agree with the venue's withdrawal policy on behalf of myself and my co-authors.